# Auxotrophic and prototrophic conditional genetic networks reveal the rewiring of transcription factors in *Escherichia coli*

Alla Gagarinova[1,2], Ali Hosseinnia[1,2], Matineh Rahmatbakhsh[1,2], Zoe Istace[1], Sadhna Phanse [1], Mohamed Taha Moutaoufik[1], Mara Zilocchi [1], Qingzhou Zhang[1], Hiroyuki Aoki[1], Matthew Jessulat[1], Sunyoung Kim[1], Khaled A. Aly [1] & Mohan Babu [1✉]

Bacterial transcription factors (TFs) are widely studied in *Escherichia coli*. Yet it remains unclear how individual genes in the underlying pathways of TF machinery operate together during environmental challenge. Here, we address this by applying an unbiased, quantitative synthetic genetic interaction (GI) approach to measure pairwise GIs among all TF genes in *E. coli* under auxotrophic (rich medium) and prototrophic (minimal medium) static growth conditions. The resulting static and differential GI networks reveal condition-dependent GIs, widespread changes among TF genes in metabolism, and new roles for uncharacterized TFs (*yjdC*, *yneJ*, *ydiP*) as regulators of cell division, putrescine utilization pathway, and cold shock adaptation. Pan-bacterial conservation suggests TF genes with GIs are co-conserved in evolution. Together, our results illuminate the global organization of *E. coli* TFs, and remodeling of genetic backup systems for TFs under environmental change, which is essential for controlling the bacterial transcriptional regulatory circuits.

[1] Department of Biochemistry, University of Regina, Regina, SK, Canada. [2]These authors contributed equally: Alla Gagarinova, Ali Hosseinnia, Matineh Rahmatbakhsh. ✉email: mohan.babu@uregina.ca

Transcription factors (TFs) are *trans*-acting DNA-binding elements utilized by the cell to modulate gene expression in response to environmental stimuli[1]. Even in the model bacterium, *Escherichia coli*, in which TFs have been widely studied, regulation mechanisms for more than half of the estimated 304 candidate TF genes remain unclear[2,3]. In *E. coli*, 14 TFs act as global regulators to modulate numerous bioprocesses[3], whereas most other TFs are local regulators for specialized bioprocess. These are encoded by genes located near target *cis*-elements, and regulated by global TFs[4]. DNA-binding sites have been elucidated by grouping co-regulated genes based on their expression levels, comparative annotations or searching for conserved sequence regions via sequence alignments[5–7]. Approaches such as DNase footprinting, chromatin immunoprecipitation (ChIP) with DNA microarray (ChIP-on-chip), deep sequencing (ChIP-seq) or modified ChIP-seq with exonuclease (ChIP-exo), and SELEX-chip have revealed DNA-binding profiles for TFs of interest[2,8–13]. Integration of disparate transcriptomics datasets from various conditions has enabled the construction of transcriptional regulatory networks in *E. coli* for 147 TFs to predict differential gene expression by environmental perturbations[14]. While these methods have provided insight into gene regulations in *E. coli*, how individual components of TF machinery are functionally coordinated, and to what extent bacterial TFs in unstressed (i.e., static) or environmental distress (i.e., differential) affect epistasis among TFs remain understudied. Consequently, we still lack contextual pathway level information for genes that lack functional annotations[2].

Here, we applied our *E. coli* synthetic genetic array (eSGA) approach[15] to interrogate static and differential genetic interaction (GI) relationships among 304 TFs of *E. coli* in auxotrophic and prototrophic culture conditions. Quantitative scoring of the resulting genetic data revealed condition-specific GIs with new components vital for the regulation of cell division, putrescine utilization (Puu) pathway, and cold shock adaptation. Further, we show interacting TF genes are co-conserved across bacterial phyla, implying conserved function in bacterial regulatory networks.

## Results

**Mapping TF static GI networks in distinct growth cultures**. To investigate the functional interconnections between TFs and their alterations due to environmental perturbation, we surveyed literature and public databases to compile a list of 232 annotated and 72 uncharacterized (or orphan) TF genes involved in diverse cellular roles (Supplementary Data 1). Static maps of auxotrophic (nutrient rich medium, RM) and prototrophic (M9 minimal medium, MM) growth conditions were constructed using eSGA[15] by generating 278 individual Hfr-Cavalli query 'donor' gene mutant strains marked with a chloramphenicol-resistance cassette, and transferred via conjugation into an arrayed 302 F-'recipient' mutant strains from Keio knockout strain collection[16], containing 295 non-essential single gene deletion, and 7 hypomorphic alleles (i.e., partial loss of gene function[15]) of essential genes marked with a kanamycin-resistance cassette constructed in house. While most (302) of the F- 'recipient' mutants grew on RM, an antitoxin (*mazE*) hypomorphic essential allele affected bacterial growth, and an alanyl-tRNA synthetase (*alaS*) Keio mutant had a partial duplication which latter suggested to be a likely essential gene candidate[17]. Due to these incongruities, both mutants were excluded from genetic screens.

After genetic transfer, double mutants were selected on RM containing both kanamycin and chloramphenicol antibiotics, and the resulting viable double mutants were replica pinned onto MM to identify GIs altered due to environmental change. The final sets of double mutant plates were digitally imaged, and growth rates

for all viable digenic mutants were determined by colony size measurements after 24 hrs in RM and MM conditions (Fig. 1a). After normalizing the colony sizes within each condition for experimental variation, and removing closely linked gene pairs (i.e., within 30 kbp on either side of the 'query' TF gene loci) with low recombination frequencies (Supplementary Fig. 1a), the average correlation of screen reproducibility between replicates was high ($r = 0.7$; Supplementary Fig. 1b) and similar to our previous GI screens[18,19].

Colony size measurements were analyzed to assign a quantitative 'static' GI ($S$) score for each digenic mutant obtained from RM ($S_{RM}$) and MM ($S_{MM}$) conditions using a multiplicative model[15]. The double mutants were also scored in parallel using our machine learning-based Gaussian process[20] to verify the GIs predicted by multiplicative model. As with our previous studies[18,19], a significant GI score was defined for each condition ($S_{RM}$, $S_{MM}$) by applying statistical thresholds corresponding to two standard deviations ($|$Z-score$| \geq 2.0$; $P \leq 0.05$) of the multiplicative ($-0.3 \leq S_{RM} \geq 1.8$; $-1.4 \leq S_{MM} \geq 3.0$) and Gaussian process ($-0.9 \leq S_{RM} \geq 1.3$; $-2.3 \leq S_{MM} \geq 1.9$) score distribution independently (Fig. 1a). The GI score reflected the extent to which a double mutant grew better (i.e., alleviating or epistasis/suppression) or worse (i.e., aggravating or synthetic sick/lethal) than the expected combination of individual mutants.

In the filtered static networks (Supplementary Data 2), over half of the digenic mutant pairs in RM (58%; 15,274 of 26,541) and MM (59%; 16,838 of 28,351) captured by both scoring models were considered as reliable GIs (Fig. 1a). For these agreement gene pairs, only GI scores from the Gaussian process were used in our analyses as the digenic mutant pairs approximated a normal distribution centered on neutrality in RM as opposed to the multiplicative model that showed a slight positive shift in the distribution of GI scores deviating from zero. Thus, from Gaussian process thresholds, we generated two static (RM, MM) genetic networks (Fig. 1b).

Comparison of significant ($P \leq 0.05$) GIs from static networks revealed many differences in RM and MM, with 45% (6587 of 14,708) of aggravating and 46% (6176 of 13,329) of alleviating GIs unique in MM not detected in RM (Supplementary Fig. 1c), indicating environmental-induced changes in epistatic relationships. As with other bacterial GI studies[18,19], static networks showed essential TFs with distinctive patterns of epistatic connectivity than non-essential TFs (Supplementary Fig. 1d). Several quality control metrics were employed to ensure data quality. This includes comparison of static TF networks to previously reported GIs that showed significant ($P = 9.0\ e^{-4}$) agreement by random sampling (Supplementary Fig. 1e). Genetically interacting TF genes within the same regulatory family or functional domain, and within the same operon in *E. coli* had positively correlated genetic profiles (Supplementary Fig. 1f, g) as pairs of co-expressed or co-transcribed TFs (Supplementary Fig. 1h, i) than random or other gene pairs.

**Static RM map reveals GI links of TF regulatory circuits**. To examine functional connections among various TF bioprocesses in static RM network, we classified 302 TF genes into 41 distinct minor processes, which were grouped into one or more of the 13 major functional categories based on literature (Supplementary Data 1). While TF genes with GIs showed intraconnectivity within each bioprocess (Fig. 1c and Fig. 2a), nearly 18% of possible non-redundant TF process combinations (254 of 1431 process pairs tested) showed significant ($P \leq 0.008$; FDR $\approx 5\%$; Supplementary Data 3) interconnectivity between bioprocesses, with over one-tenth displaying aggravating (~10%; 148 of 1431) and alleviating (~12%; 165 of 1431) interactions.

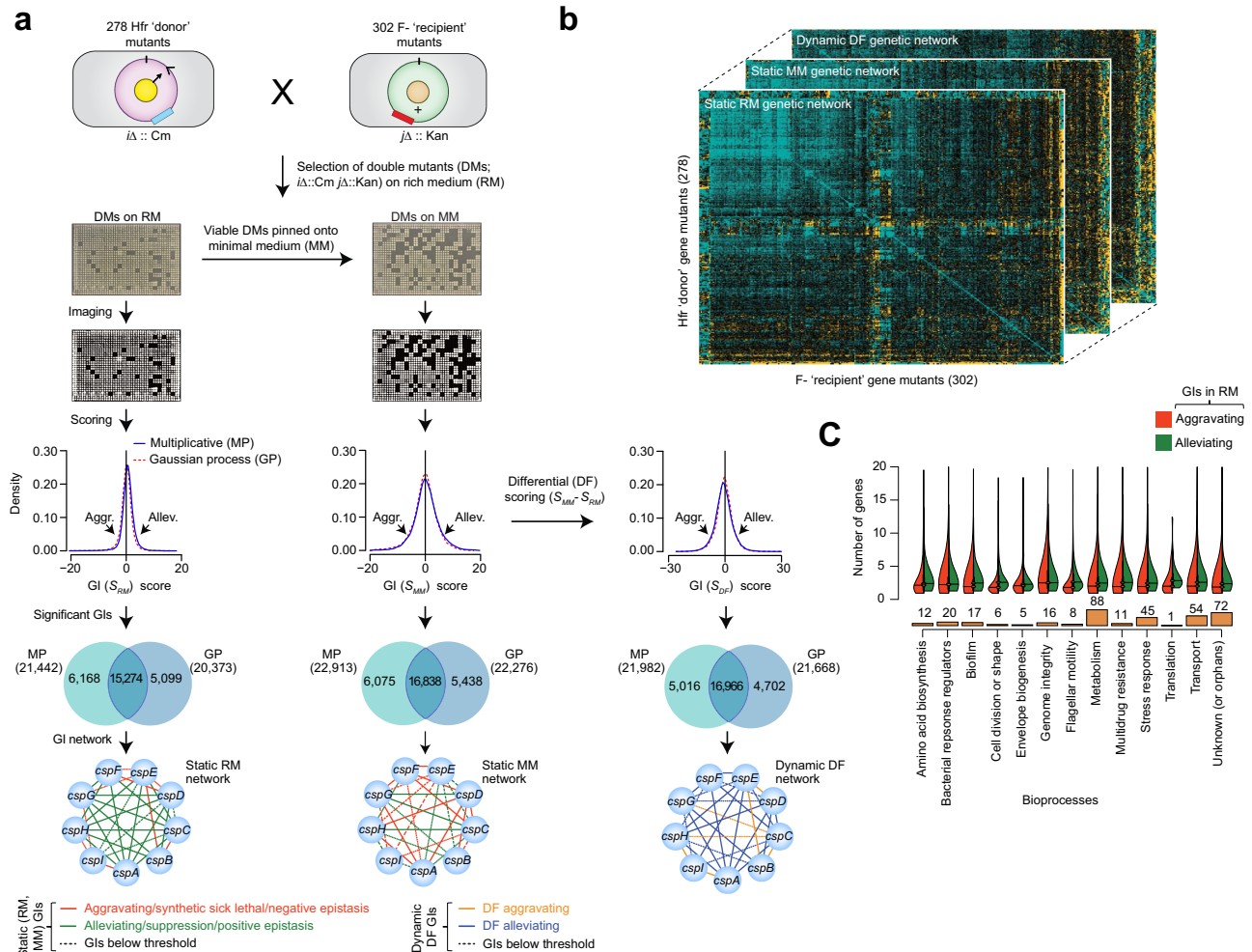

**Fig. 1 TF gene targets and static and differential GI screening pipeline. a** Schematic of the conjugation, double mutant generation in RM and MM growth conditions, colony imaging, mutant strain growth rate measurements ($n = 8$ replicate gene pairs per donor) using multiplicative and Gaussian process models, and construction of static and differential genetic networks using the significant GI scores derived from respective growth conditions. **b, c** TF genes screened as donors and recipients (**b**) to generate static and differential networks, and their assignment (**c**) into 13 broadly representative bioprocesses. Source Data are provided as a Source Data file.

Under detailed inspection, the static map coincided with known functional dependencies (Fig. 2a and Supplementary Fig. 2a). For example, alleviating GIs between cold shock and stress-responsive TF genes reveal cooperative role in the regulation of stress response[21]. Conversely, aggravating GIs include connections between TFs of two-component signal transduction and motility or biofilm, reflecting two-component regulators in controlling motility or biofilm[22]. In some cases, unexpected functional dependencies suggested new hypotheses (Supplementary Fig. 2a). For example, coordination of TFs of acid resistance (AR) with cellular metabolism were enriched ($P = 5.2\,e^{-3}$; Supplementary Data 3) for alleviating GIs, which may implicate a regulatory network interconnecting these processes[23].

Enrichment ($P = 3.1\,e^{-2}$) for aggravating GIs was also observed between metabolism and flagellar motility (Supplementary Fig. 2a), suggesting that impairment of metabolic activities required for flagellar assembly may result in the failure of motility functions[24]. Also, as with RM network (Supplementary Data 4), positively correlated GI profiles among essential genes in antitoxin system (*chpS, mqsA, yefM*) and SOS (*dnaA, lexA*) regulon is consistent with single gene mutant resistance profiles to antibiotics from our essential phenomics screens (Supplementary Fig. 2b), supporting the possible function of a SOS-regulated toxin-antitoxin system[25].

The static RM map further revealed the regulatory modules involving gene pairs where one TF represses the function of another TF that acts as an activator (Fig. 2b-1), or two different TFs that function cooperatively (Fig. 2b-2) or redundantly (Fig. 2b-3) to regulate a distinct set of genes. We have discussed each of these relationships for literature-supported regulatory gene pairs by associating with their GI patterns. An example of the first case (Fig. 2b-1) is the repression of a TF gene encoding the cyclic AMP receptor, *crp*, by the nucleoid-associated protein (NAP), Fis. This implies inhibiting *crp* in *fis* mutant should display alleviating GI. As a global regulator, *crp* activates several genes, including *cspE*, a member of the *cspA* family of cold shock genes in *E. coli*. Since *cspE* and *crp* play a role in cold adaptation, and deletion of *crp* leads to a growth defect at low temperature[26], loss of *crp* and *cspE* alleles at optimal temperature should manifest as alleviating GIs, and we in fact detected this relationship. Notably, *fis* and *cspE* mutants resulted in aggravating GI, implying their joint influence on stress-response regulation[27].

In the second case (Fig. 2b-2), aggravating GI between the NAP encoding gene, *hns*, and its paralog, *stpA*, that repress each other transcriptionally[28], supports cell growth impairment observed upon *stpA* inactivation in *hns* mutant[29]. However, since *stpA* activates or represses H-NS regulated genes[28], we asked if *stpA*

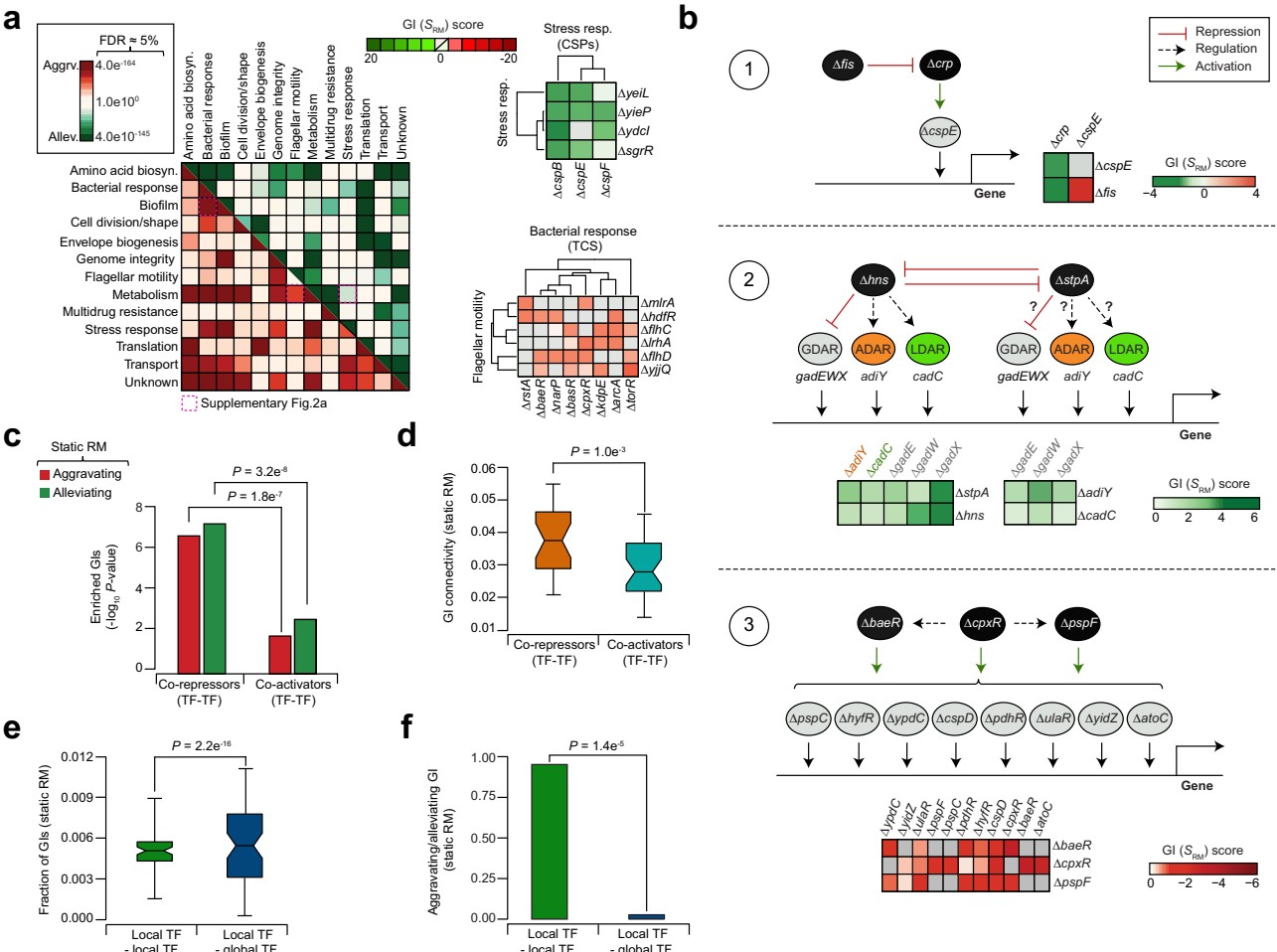

**Fig. 2 Epistatic patterns/connectivity of TFs in static RM network. a** Enrichment (*P*-value by one-tailed hypergeometric test, and adjusted using the Benjamini-Hochberg FDR correction) of crosstalk among bioprocesses (*n* = 254 of 1431 process pairs tested) with distinct aggravating or alleviating GI patterns in RM is shown along with illustrative examples (see also Supplementary Fig. 2a). **b** Representative regulatory modules involving TF gene pairs as regulators, activators, or repressors of gene expression, and their respective epistatic relationships are shown. Since StpA and H-NS have overlapping and distinct functions, we predict (indicated in question mark) that like *hns*, *stpA* may repress or regulate acid resistance (AR) genes based on the alleviating GI patterns with AR targets. **c**, **d** Enrichment of aggravating (*P* = 1.8 e$^{-7}$) or alleviating (*P* = 3.2 e$^{-8}$) GIs (**c**; from left to right, *n* = 17 and 27 gene pairs; *P*-value by hypergeometric test), and epistatic connectivity (**d**; from left to right, *n* = 27 and 35 genes) between TF genes that function as co-repressors or co-activators (*P* = 1.0 e$^{-3}$ by two-sided Wilcoxon signed-rank test). Box plot (**d**) from left to right is shown with maximum (0.055 and 0.044), minimum (0.021 and 0.013), quartile 1 (Q1; 0.029 and 0.022), Q2 (0.037 and 0.027) and Q3 (0.047 and 0.035) values. **e**, **f** GI frequency (**e**; from left to right, *n* = 182 and 182 genes; *P* = 2.2 e$^{-16}$ by two-sided Wilcoxon signed-rank test), and ratio of aggravating to alleviating GIs (**f**; from left to right, *n* = 182 and 182 genes; *P* = 1.4 e$^{-5}$ by two-sided Wilcoxon signed-rank test) observed between local TFs compared to local with global TF gene pairs. Box plot (**e**) from left to right is shown with maximum (0.009 and 0.010), minimum (0.0015 and 0.00), Q1 (0.0041 and 0.003), Q2 (0.005 and 0.0055) and Q3 (0.0058 and 0.008) values. Source data are provided as a Source Data file.

regulates *hns* gene targets, such as arginine (*adiY*), glutamate (*gadEXW*) and lysine (*cadC*)-dependent AR (ADAR, GDAR, LDAR)[30], then overcoming *hns*/*stpA* effect on AR systems can be achieved by deleting *adiY*, *gadEXW* or *cadC*, resulting in an alleviating phenotype. Indeed, our GI map identified these patterns, and bridged ADAR, GDAR and LDAR systems, suggestive of crosstalk between interrelated AR regulatory networks[23], and could represent a situation where ADAR, GDAR or LDAR, and *hns*/*stpA* act in a coordinated manner.

The third case (Fig. 2b-3) is related to redundant envelope stress response regulons (i.e., Bae, Cpx, Psp, Rcs, σ$^E$), comprising overlapping gene targets of *baeR* and *pspF* that are regulated by another TF, *cpxR*[31]. Our static map unveiled an aggravating GIs among *baeR*, *cpxR* and *pspF* mutants similar to a previous report[32], suggesting functional crosstalk of the TF gene targets regulated by CpxR and other envelope stress response signaling systems.

Next, we explored whether the regulatory connection of TFs and their cognate targets from RegulonDB database can be linked to GIs observed within the RM network. We found many TFs and their cognate target TFs that were co-repressed are enriched for aggravating (*P* = 1.8 e$^{-7}$) or alleviating (*P* = 3.2 e$^{-8}$; Fig. 2c) GIs when compared to pairs that function as co-activators. This observation is in support of the one-third (34%, 17 of 50) co-repressed TF pairs that displayed high GI connectivity (*P* = 1.0 e$^{-3}$; Fig. 2d) in static RM compared to interactions from over half of the TF co-activators (59%, 27 of 46). These results suggest that in contrast to activators, repressors co-evolve tightly with their cognate targets[33], leading to disparate GI connectedness.

To investigate whether GIs between TF gene pair can be explained indirectly by interactions between their targets, we analyzed GIs of the predicted targets from RegulonDB database for each pair of TFs. As in yeast[34], TF pairs with strong GIs were

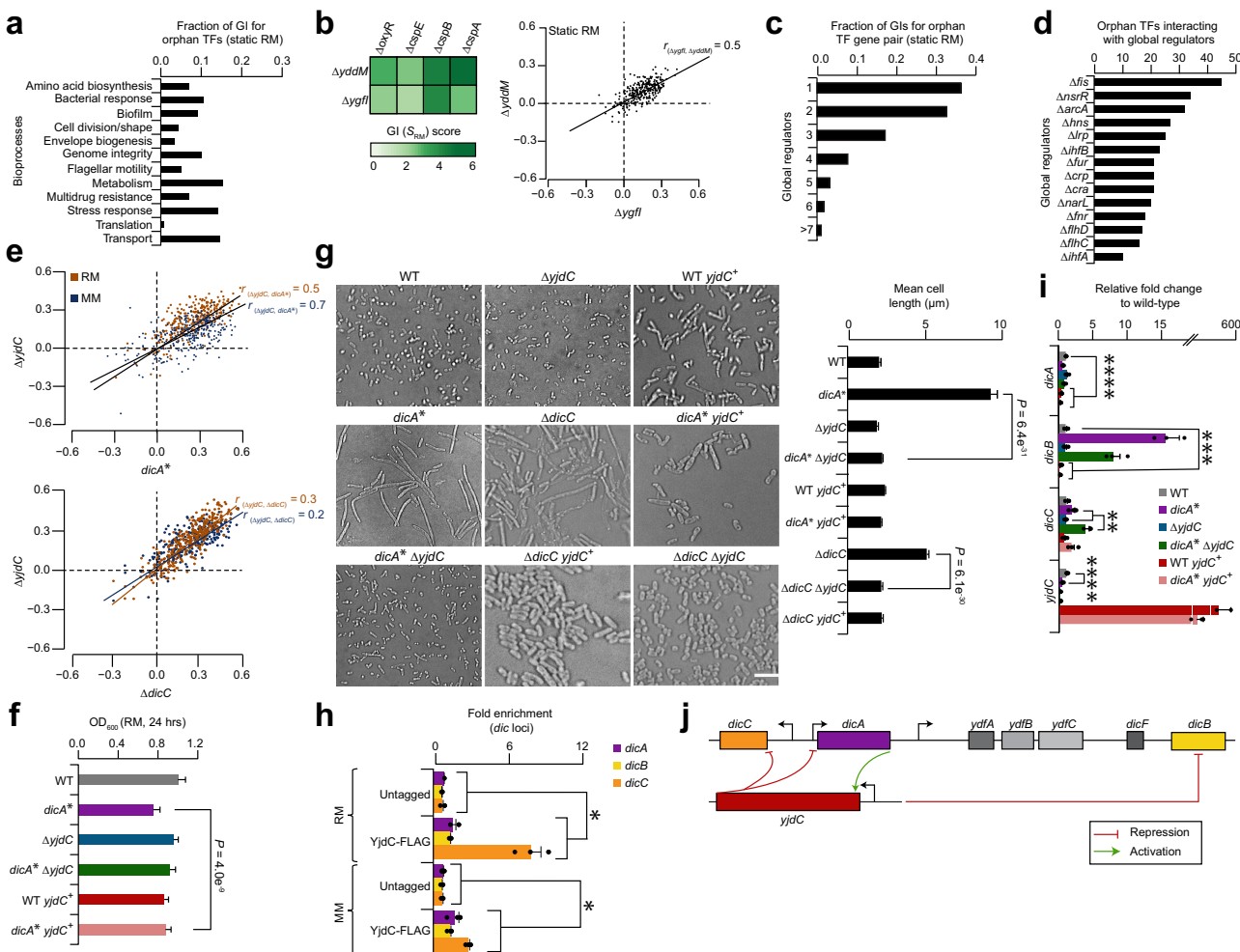

**Fig. 3 Orphan TF epistatic connections in static RM network. a** Frequency of orphan TF interactions with annotated TF genes in the indicated bioprocesses. **b** Sub-network of two orphan TF genes (*yddM, ygfI*) interacting with stress response regulators, and their correlated GI profiles ($n = 300$ gene pairs). **c** Distribution of orphan TF gene pairs ($n = 1913$) interacting with one or more global regulators. **d** Target global regulators, and their GI frequency with orphan TF genes. **e** Correlated GI profiles between *yjdC* and *dicA* or *dicC* in RM and MM (from top to bottom, $n = 300$ gene pairs for RM and MM, respectively). **f, g** Growth rate (**f**; $n = 12$ biologically independent experiments; $P = 4.0 e^{-9}$ by Student's two-sided *t*-test at 32 °C with $OD_{600}$ measured at 24 h, and representative cell morphology micrographs ($T = 4$ h after reaching OD = 0.5–0.6) along with cell length measurements (**g**; $n \geq 25$ cells over five biologically independent experiments; $P = 6.4 e^{-31}$ and $6.1 e^{-30}$ by Student's two-sided *t*-test) of essential *dicA* hypomorph (*) or *dicC* mutant and *yjdC* double or single mutants, or overexpression of *yjdC* in wild-type (WT), *dicC* mutant or *dicA* hypomorph in RM. Scale bar represents 10 μm. **h** Fold enrichment of YjdC binding to *dicABC* genes by ChIP-qPCR in RM and MM ($n = 3$ biologically independent experiments) was analyzed by comparing chromosomal YjdC FLAG₃-tag to untagged control strain, and normalized to a negative control (i.e., primers designed away from YjdC binding site of *dic locus* was used to amplify from YjdC-FLAG₃). **i** Transcript levels of *yjdC* and *dic* genes relative to WT measured by qRT-PCR ($n = 3$ biologically independent experiments) in the strains indicated. Significance in panels **h** and **i** (*$P \leq 0.05$, **$P \leq 0.01$, ***$P \leq 0.001$, **** $P \leq 0.0001$) are calculated using Student's two-sided *t*-test. **j** Model illustrating the potential role of YjdC in cell division. Data (**f–i**) are presented as mean ± standard deviation from the indicated number of independent samples. Source data are provided as a Source Data file.

not enriched for GIs between their targets (Supplementary Fig. 2c). One-third of TF pairs (32%, 8200 of 25,749) from RM and MM networks that exhibited significant GIs have at least one pair of targets with a stronger GI, which is lower than random (37%; 9500 of 25,749). Thus, indirect effects of the targets are not the likely reason for the large number of GIs observed between TFs.

When we assessed the degree to which bioprocesses were enriched in GIs involving global or local TF regulators, we found enrichment (FDR ≈ 1%) for one-third (31%; 4 of 13) of all targeted bioprocesses (Supplementary Fig. 2d and Supplementary Data 5), comprising a local TF that is interacting with a global TF than with another local regulator ($P = 2.2 e^{-16}$; Fig. 2e), as reported for yeast[34]. While a majority of local TFs

detected for aggravating GIs appear to function in a redundant manner, significantly ($P = 1.4 e^{-5}$) more alleviating GIs were observed among the global and local TF gene pairs (Fig. 2f), indicating that local TFs work together with a global regulator[4,34]. Strikingly, bioprocesses related to local TFs of metabolism interacted with other local and global regulators of bacterial response, genome integrity, and transport (Supplementary Fig. 2d). For example, local regulatory metabolism genes, such as *ebgR* (β-galactosidase repressor), and *appY* (acid phosphatase) or *araC* (arabinose regulator) of GalR-LacI and AraC-XylS families, respectively, showed GI on average with 35 other local and 4 global regulators, reflecting the genetic dependencies between TFs and the target genes they regulate.

**Static RM map shows functional connections for orphan TFs.** Since epistatic connections remain unknown for about a quarter (24%, 72 of 304) of *E. coli* unannotated TFs, we examined the RM network to find functional roles for these TFs based on their GIs with annotated genes. Our analysis revealed a sub-network of 5921 high-confidence ($|Z\text{-score}| \geq 2.0$; $P \leq 0.05$) GIs involving an orphan and annotated gene (Supplementary Data 2), reflecting the functional crosstalk of orphan TFs with bioprocesses. However, the GIs identified per orphan TF varied considerably (Supplementary Data 6), ranging from 47 (e.g., *yqeI*) to 129 (e.g., *ycaN*). Over one-fourth (28%; 655 of 2346) of orphan TF pairs showed interconnectivity with annotated genes of various enriched ($|Z\text{-score}| \geq 2.3$; $P \leq 0.01$) bioprocesses (Supplementary Data 7), of which one-quarter (24%, 158 of 655) shared with metabolism, transport, and stress response genes (Fig. 3a). This finding is consistent with orphans connected to annotated TFs enriched (FDR ≈ 1%) for these processes (Supplementary Data 6). For example, as with stress regulators (*cspABE*, *oxyR*) shared between *ygfI* and *yddM* by alleviating GIs (Fig. 3b), the genetic profiles of these orphans were also correlated ($r_{ygfI, yddM} = 0.5$), suggestive of their joint function with stress response genes.

Examination of orphan genes displaying GIs with global regulators indicated that over one-third (39%, 745 of 1913) of orphan TF pairs have increased GIs with a global regulator as opposed to very few interactions (1%, 16 of 1913) with many (≥7) different global regulators (Fig. 3c). In support of this, each global regulator revealed varying degrees of GIs with orphan TFs (Fig. 3d). For example, the NAP encoding gene, *fis*, showed strong GI with two-third (65%; 45 of 69) of the orphan genes, while the other NAP gene, *ihfA*, interacted only with 10 different orphan TFs. These results suggest that while GIs with very few global regulators follows a power-law distribution[4], most orphan TF pairs are rarely targeted concurrently by many global TFs, or exhibit low tendency of cooperative or redundant regulation by multiple global regulators.

Given that 304 candidate TFs were classified into 73 different families as per RegulonDB database (Supplementary Data 1), we explored the functional dependencies between TF families of an orphan and annotated genes involved in an enriched bioprocess based on GI connectivity. Nearly one-tenth (8%; 79 of 1007) of all non-redundant family pairs tested showed interconnected GIs between an orphan and annotated gene of two different TF families that were significantly ($P \leq 0.05$) enriched for bacterial bioprocesses such as metabolism, envelope biogenesis, cell division/shape, and flagellar motility, among others (Supplementary Fig. 2e). For example, we were intrigued by the alleviating GI between the unannotated gene *yjdC*, a member of TetR/AcrR family, and cell division inhibition gene, *dicA*, of helix-turn-helix (HTH) family (Supplementary Data 1) that showed positively correlated GI profiles ($r_{yjdC, dicA} = 0.5$ in RM or 0.7 in MM; Fig. 3e) and ranked top 1% among all correlated genetic profiles in RM network. This functional dependency was confirmed by liquid growth assays (Fig. 3f), where unlike *yjdC* mutant, *dicA* hypomorphic allele showed reduced growth in RM, and overexpression of *yjdC* rescued the growth defect of *dicA* mutant in a similar manner to *yjdC dicA* double mutant.

To gain further insight into the functional connection between *yjdC* and *dicA*, we relied on prior observations that the *dic* locus-containing Qin cryptic prophage-encoded *dicA* represses *dicB* and *dicC*, while concurrently activating its own expression[35]. The *dic* locus is also transcriptionally controlled by *dicC*, and a temperature sensitive *dicA* allele derepresses *dicB* and *dicC*, prompting a filamentous phenotype[35]. In vitro analyses imply *dicA* may bind to the operator sequence of *dicC*, likely in the presence of another unknown TF[36]. We posit that *yjdC* regulates the expression of *dic* locus in a joint manner. To assess this, we

examined the bacterial morphology in *yjdC* and *dicA* mutants, as well as between *yjdC* and *dicC* that showed modest positive correlation ($r_{yjdC, dicC} = 0.3$ in RM or 0.2 in MM; Fig. 3e) in static networks. Unlike *yjdC*, *dicA* or *dicC* mutant as expected[35,37] formed long filamentation (~9.4 µm to ~5.2 µm avg. cell length). Strikingly, the *dicA* or *dicC* phenotype was rescued to near wild-type (~2.2 µm) only when *yjdC* was additionally deleted (Fig. 3g). Overexpression of *yjdC* in trans in wild-type, and *dicA* or *dicC* mutant exhibited filamentation, albeit to reduced cell length (~2.2 to ~2.4 µm) than *dicA* or *dicC* single mutant.

We reasoned the suppression of filamentous growth of *dicA* or *dicC* mutant by *yjdC* deletion mutant could be due to *dic* genes transcriptionally regulated by *yjdC*. To test this, ChIP-qPCR was performed to ascertain whether YjdC binds to *dic* locus. In comparison to untagged control strain, chromosomal FLAG$_3$-tagged YjdC grown in RM and MM showed a 2.0- to 12.0-fold enrichment of YjdC occupancy at *dicABC*, respectively (Fig. 3h), implying *dic* alleles are likely targets of YjdC. Measurement of *dic* transcript levels in *yjdC* mutants by quantitative real-time PCR (qRT-PCR) further led to four vital observations (Fig. 3i). Firstly, while in *dicA* hypomorphic allele, as expected, the *dicA* transcript level was reduced, we noticed both *dicB* and *dicC* transcripts to be concurrently increased above wild-type, with *dicC* expression reduced to one-tenth the *dicB* level. This finding implies increase of *dicB* transcript may mediate the filamentation of *dicA* mutant, consistent with *dicA* inhibiting *dicB* expression to promote cell division[38]. Secondly, in contrast to wild-type, a 3-fold decrease of *yjdC* transcript in *dicA* mutant suggested *dicA* as an activator of *yjdC*. Thirdly, in contrast to *dicB*, there was a 2.0-to-4.4-fold increase in *dicC* transcript level when *yjdC* was deleted in *dicA* hypomorph than *dicC* expression in the respective single mutant strains, implying that *yjdC* acts as a repressor of *dicC* when *dicA* is impaired.

Fourthly, while *dic* transcript levels were not significantly affected in *yjdC* mutant compared to wild-type, overexpression of *yjdC* in a wild-type or *dicA* mutant led to a 3-to-12-fold reduction in *dicAB* transcript levels than wild-type, with no significant impact on *dicC* expression. This suggests that *yjdC* downregulates *dic* locus when expressed in trans, while *dicA* largely controls *dic* locus under wild-type conditions. Together, these data support a model (Fig. 3j) in which *dicA* activates *yjdC*, *yjdC* represses *dicC* in a *dicA* null allele, and *yjdC* act as a repressor of *dicA* and *dicB* when *yjdC* is overexpressed. To ensure that the control of *dic* locus does not irreversibly switch to *dicABC* regulation, *yjdC* binds to *dic* locus, facilitating *dicABC* expression to resume cell division. Based on these findings, we rename *yjdC* as *dicD* (for division control) as it acts as a repressor by controlling the transcription of *dic* genes in cell division process.

**Static maps accentuate YneJ in regulating Puu pathway.** Another orphan TF gene of unknown function in our static networks was the LysR-family transcriptional regulator, *yneJ*, which showed strong alleviating GI with an HTH-type transcriptional regulator *PuuR* that regulates transcription of *puu* genes in Puu pathway[39] (Supplementary Data 2). This gene pair that ranked top 7 among their other pairwise alleviating GIs in RM was confirmed by liquid growth assays (Fig. 4a), suggesting a cooperative role in putrescine catabolism. The *yneJ* gene is located close to the sigma E promoter region of *yneI* operon, encoding succinic semialdehyde dehydrogenase (*yneI* or *sad*), glutaminase (*yneH* or *glsB*), and a DUF4186 domain-containing orphan, *yneG*, of unknown function (Fig. 4b). In addition to their constitutive pathway, *E. coli* use L-glutamine (Gln), putrescine, or other metabolites as carbon or nitrogen sources to increase their survival via the Puu pathway, which are triggered during carbon or nitrogen limiting conditions[40].

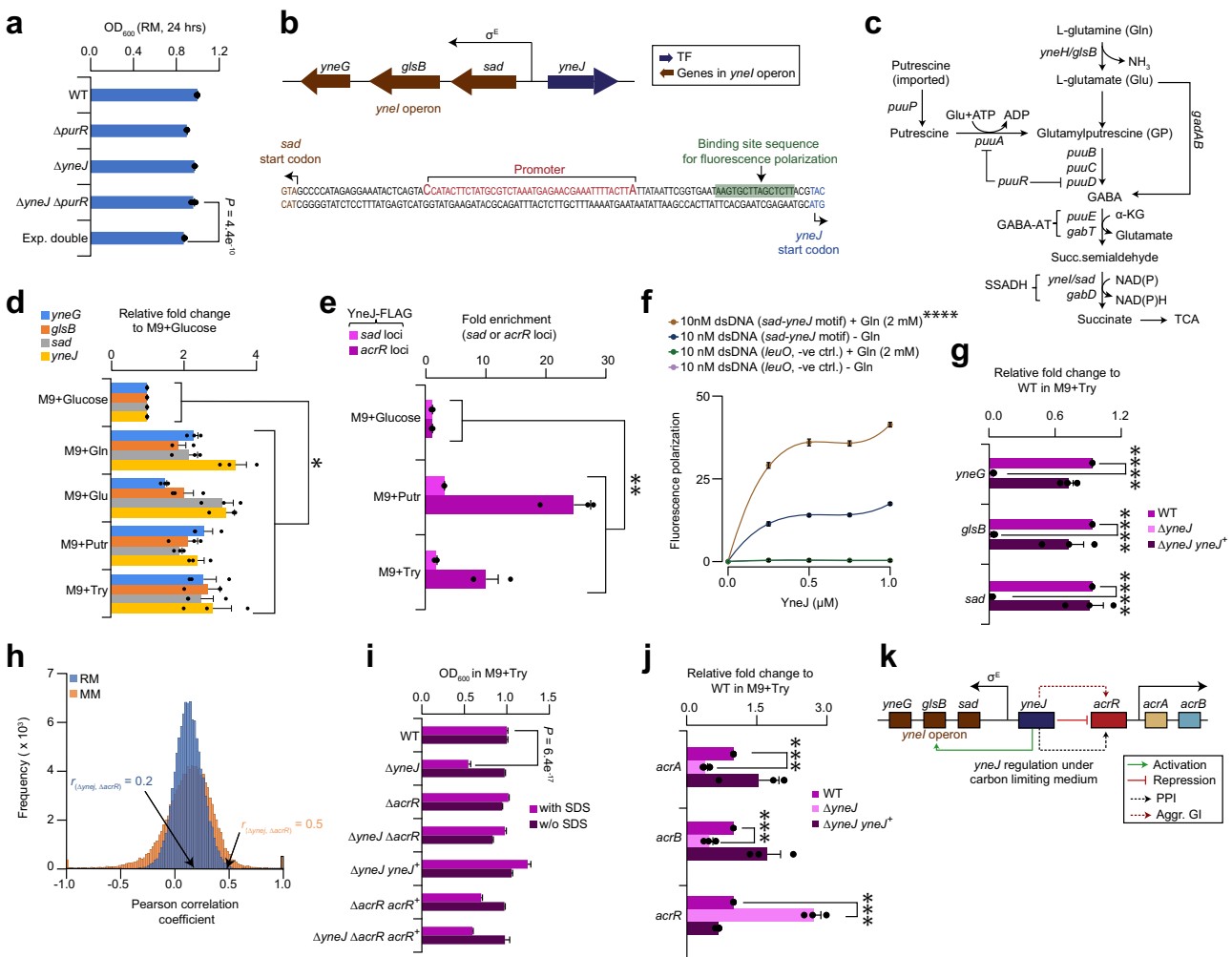

**Fig. 4 YneJ regulates the components of putrescine pathway and efflux pump. a** Growth rate of indicated strains at 32 °C with $OD_{600}$ measured ($n = 10$ biologically independent experiments; $P = 4.4\,e^{-10}$ by Student's two-sided $t$-test) at 24 h in RM. **b** Location of *yneJ* near the $\sigma^E$ promotor region of *yneI* operon, comprising *sad-glsB-yneG* genes, with DNA binding site (highlighted in green) predicted within the *sad-yneJ* intergenic region based on conservation analysis. **c** Genes in putrescine utilization and Gln-Glu pathways. **d** Transcript levels of indicated genes in a wild-type (WT) BW25113 strain grown in M9 media with different carbon sources (0.5% Gln, Glutamine; 0.5% Glu, Glutamate; 0.4% Putr, Putrescine; 1% Try, Tryptone) relative to 0.1% M9-glucose ($n = 3$ biologically independent experiments). **e** Fold enrichment of YneJ binding site in *sad-yneJ* or *acrA-acrR* intergenic region measured by ChIP-qPCR ($n = 3$ biologically independent experiments) in M9-putrescine, tryptone and glucose media. ChIP enrichment in a chromosomal YneJ FLAG$_3$-tag normalized to a negative control (i.e., primers designed away from YneJ binding site of *sad* or *acrR* was used to amplify from YneJ-FLAG$_3$). **f** YneJ binding to *sad-yneJ* site by fluorescence polarization ($n = 4$ biologically independent experiments) is shown at increasing concentration of YneJ recombinant protein with 10 nM fluorescence-labeled double-stranded DNA (i.e., *sad-yneJ* consensus motif or promoter region of *leuO* gene that serve as negative control), and in the presence or absence of Gln (2 mM). Significance (****$P = 1.7\,e^{-9}$) calculated using Student's two-sided $t$-test between *sad-yneJ* consensus motif with Gln vs. without Gln at 1 μM concentration of YneJ protein. **g** Transcript levels of target genes measured by qRT-PCR ($n = 3$ biologically independent experiments) in *yneJ* mutant or complemented (+) strains vs. WT grown in M9-tryptone. **h** Distribution of correlation coefficient profiles among TFs, with emphasis on *yneJ-acrR* GI profile ($n = 300$ gene pairs) in RM and MM. **i** Growth rates of WT and indicated mutant or complemented (+) strains in M9 media containing tryptone, with or without SDS ($n = 12$ biologically independent experiments; $P = 6.4\,e^{-17}$ by Student's two-sided $t$-test) at 32 °C over 24 h. **j** Changes in transcript levels of target genes in *yneJ* mutant or complemented (+) strain vs. WT grown in M9-tryptone ($n = 3$ biologically independent experiments). **k** Model illustrating YneJ role in the regulation of Puu and efflux pump. Data (**a**, **d–g**, **i**, **j**) are presented as mean ± standard deviation from the indicated number of independent samples. Significance in panels **d**, **e**, **g** and **j** (*$P \leq 0.05$, **$P \leq 0.01$, ***$P \leq 0.001$, ****$P \leq 0.0001$) calculated by Student's two-sided $t$-test. Source data are provided as a Source Data file.

Typically, Gln, upon uptake into *E. coli* is converted to L-glutamate (Glu) by glutaminase gene, *glsB*. The Glu is then decarboxylated to γ-aminobutyric acid (GABA) by Glu decarboxylase, *gadAB*, and feeds into the Puu pathway (Fig. 4c), where putrescine imported through *puuP* is γ-glutamylated by *puuA*. The ensuing γ-glutamylputrescine to GABA by *puuBCD* is transferred by GABA aminotransferase (*puuE, gabT*) to succinic semialdehyde, which is oxidized by succinic semialdehyde dehydrogenase (*sad, gabD*) to yield succinate for the tricarboxylic

acid cycle[40]. The *puuR* TF gene that is also located in the *puu* gene cluster functions as a repressor of *puuAD* genes[39]. While *glsB-sad* are vital in ammonium assimilation and putrescine utilization[40,41], *yneJ* role in regulating *yneI* operon in Gln-Glu and Puu pathways remains unclear. We posit that *yneJ* regulates *sad-glsB-yneG* genes of *yneI* operon by using Gln/Glu/putrescine/tryptone as a sole carbon source.

To test our hypothesis, *yneJ* mutant growth was compared to wild-type *E. coli* strain in M9 carbon-limiting medium

containing, Gln, Glu, putrescine or tryptone as a lone carbon source, and $NH_4Cl$ as a nitrogen source. Under M9-tryptone condition, *yneJ* mutant grew similar to wild-type, but barred in M9-Gln/Glu/putrescine. Ectopic expression of *yneJ* in *yneJ* mutant stimulated growth to wild-type levels in all culture conditions tested (Supplementary Fig. 3a). This suggests that YneJ is required for putrescine catabolism, and since *sad* is putrescine inducible[40], and Gln/Glu are precursors of putrescine, it is likely that *yneJ* mutant is unable to regulate *sad*, thereby failing to utilize Gln/Glu/putrescine, resulting in the growth defect.

Given that *sad* expression is influenced by carbon or nitrogen limiting culture conditions, as *puu* gene cluster[40], we measured the transcript level of *yneJ* and genes within the *yneI* operon in a wild-type strain grown in M9-Gln/Glu/putrescine/tryptone medium vs. M9-glucose by qRT-PCR. Remarkably, *sad-glsB-yneG* and *yneJ* expression level was elevated from 1.5- to 3.4-fold in M9-Gln/Glu/putrescine/tryptone than in M9-glucose medium (Fig. 4d), indicating as with Sad[40], YneJ utilizes Gln/Glu/putrescine/tryptone as a lone carbon source.

To ascertain YneJ binding under carbon-limiting conditions, a ChIP-seq was performed using a YneJ $FLAG_3$ chromosomal tag at C-terminus in DY330 strain grown in M9-putrescine/tryptone medium. After stringent peak-calling analysis ($P \leq 1.0\,e^{-4}$), genes corresponding to the start codons within 200 bp upstream and downstream of a binding site were considered as putative targets. This resulted in 11 YneJ-binding sites in M9-putrescine and 7 in M9-tryptone media (Supplementary Data 8). These unreported binding sites were upstream, inside, or in the intergenic regions of small RNAs (e.g., GadY, an activator of *gadAB* of Glu decarboxylase) or transfer RNA genes, which have regulatory roles in gene expression during nutrient constraint[42,43]. While barely missing the threshold in peak calling analysis, YneJ binding sites were found upstream of *sad*, *glsB* and *yneG* in M9-putrescine and/or tryptone media, consistent with binding properties varying with growth conditions[11]. To ensure the *sad* regulatory region is a genuine YneJ binding site and not from noise, we performed ChIP-quantitative PCR (ChIP-qPCR) on an endogenous $FLAG_3$-tagged YneJ in M9-tryptone/putrescine media, revealing a 1.8- to 3.2-fold enrichment of YneJ occupancy upstream of *sad* (i.e., *sad-yneJ* intergenic region; Fig. 4e) compared to YneJ-FLAG control strain grown in M9-glucose.

We next inquired whether the *sad* regulatory region recognized by YneJ can be inferred from conservation based on the presence of *yneI* operon and YneJ orthologs among γ-Proteobacterial species. Phylogenetic analysis on the orthologs of *yneI* operon and *yneJ* across 54 closely related γ-proteobacteria indicated that nearly half (43%; 23 of 54) of the species had a high sequence co-conservation ($P = 2.1\,e^{-3}$; Supplementary Fig. 3b). Alignment of DNA sequences of *yneI* operon and *yneJ* to γ-Proteobacterial species further revealed three enriched ($E$-value = 0) motifs by MEME suite (Supplementary Fig. 3c, d). Two (TTCTCGATTCGTGAA, AATATTATTCATTTT) were found within the promoter region of *yneI* operon, while the third (CCAGTTGGGTCAGAT) resided within the *yneJ* gene. To ascertain whether the identified motif is the genuine YneJ binding site for *sad* regulation, we performed fluorescence polarization assay with 21-bp fluorescent-labeled oligonucleotides as DNA probes designed for one (TTCTCGATTCGTGAA, Fig. 4b) of the motifs as it was consistent with ChIP-qPCR targeting *sad-yneJ* region (Fig. 4e).

Since Gln serves as a precursor for putrescine synthesis and in the generation of metabolites (i.e., Glu production from Gln via *glsB*), incubating purified YneJ-$His_6$ recombinant protein (Supplementary Fig. 3e) with Gln/Glu/putrescine should enhance its binding to the *sad* regulatory region. Indeed, this was the case as polarization increased with increasing concentration of YneJ

protein with the addition of Gln (i.e., 2 mM, 100 μM, 2 μM, 100 nM, 25 nM, 2 nM) and Glu or putrescine (2 mM, 2 μM) at various dilutions, and bound to the probe more efficiently (data shown for 2 mM Gln, Glu or putrescine; Fig. 4f and Supplementary Fig. 3f) than YneJ without the metabolites or with *leuO* negative control gene. The efficiency of YneJ binding to *sad* was also more robust with Gln than Glu or putrescine. The result suggests that while YneJ binds to the conserved site in *sad-yneJ* intergenic region with Glu or putrescine, albeit at high affinity with Gln, the site is likely utilized by YneJ for regulating the expression of *sad-glsB-yneG* genes of *yneI* operon. To confirm the latter, mRNA expression of *sad-glsB-yneG* was examined by qRT-PCR in wild-type and *yneJ* mutant grown in M9-tryptone medium. We found *sad-glsB-yneG* mRNA levels (Fig. 4g) were lower in *yneJ* mutant, while *yneJ* overexpression in *yneJ* mutant restored mRNA transcript near to wild-type. These results confirm that *yneJ* is an activator of *yneI* operon when utilizing Gln/Glu/tryptone/putrescine as the sole carbon and vital nutrient source.

To understand whether the isozymes in Gln-Glu and Puu pathways were also regulated by *yneJ* besides *sad-glsB-yneG*, we examined the ChIP-seq data, and found YneJ to bind upstream of Glu decarboxylases (*gadAB*), *puu* gene cluster (*puuABCDR*), GABA aminotransferases (*puuE*, *gabT*), and succinic semialdehydes (*sad*, *gabD*) in M9-putrescine and/or tryptone media just below the ChIP-seq peak-calling analysis threshold. However, affinity-purified, endogenously $FLAG_3$-tagged YneJ when grown under Luria-Bertani (LB) and M9-putrescine/tryptone medium, followed by mass spectrometry, co-purified with members (e.g., GlsB, GadB, PuuACE, GabT, Sad, GabD) of Gln-Glu and Puu pathways (Supplementary Fig. 4a). The mRNA levels for genes in this pathway, except the *yneI* operon and *gabD*, measured by RNA-sequencing (RNA-seq) were higher in *yneJ* mutant than wild-type grown in M9-tryptone (Supplementary Fig. 4b). This implies *yneJ* regulates Puu pathway genes by activating *sad-glsB-yneG* or *gabD* expression, while repressing others. Since *yneJ* (Supplementary Fig. 3a) or *sad* mutant exhibit growth defect in putrescine but not *gabD* mutant[40] suggests that *sad* is vital than *gabD* when *E. coli* utilizes putrescine as a carbon source[40], and that *yneJ* seems to explicitly regulate *yneI* operon in putrescine-inducible pathway.

**Static maps links *yneJ* with *acrR* in Puu pathway.** Besides characterizing the physiological role of YneJ, we found *yneJ* exhibiting strong aggravating GIs (Supplementary Data 2) and similar genetic profiles (Fig. 4h) with a multidrug efflux regulator, *acrR*, in both RM and MM conditions. This epistatic connection was probed as it ranked top 2 in MM network among all *yneJ* correlated genetic profiles, and confirmed by liquid growth assays (Supplementary Fig. 4c). We sought to comprehend how the aggravating GI of *yneJ* with *acrR* is linked to Puu pathways. In response to shifts in nutrient availability, the transcriptional regulator *acrR* acts as a local repressor of *acrAB* efflux pump by fine tuning *acrAB* transcription and averting excessive *acrAB* expression, which otherwise could lead to a loss of membrane potential that is deleterious to *E. coli*[44]. While *acrR* senses many toxic metabolites and changes in environmental conditions, inhibition of *acrAB* by *acrR* would amass the metabolites. Thus, halting *acrR* sparks *acrAB* initiation to increase bacterial fitness under stress[44]. These observations led us to posit that *acrR* modulation by *yneJ* can induce *acrAB* expression to extrude toxic metabolite remnants from Puu pathway.

To test the hypothesis, we performed co-immunoprecipitation experiments to ascertain whether *yneJ* interacts physically with *acrR*. YneJ strain carrying a chromosomal $FLAG_3$ tag co-

precipitated with His$_6$-tagged AcrR in RM and M9-tyrptone conditions (Supplementary Fig. 4d). Further confirmation of this interaction by bacterial two-hybrid assay (Supplementary Fig. 4e) implies that physical association between these TF regulators may be vital for the modulation of AcrAB activity. As with yeast studies[45], the *yneJ-acrR* gene pair whose products physically interact and share an aggravating GI further suggests that they possess a functional relationship. Because *acrR*-deficient strain confers resistance to sodium dodecyl sulfate (SDS) by increasing AcrAB activity[44], we challenged both *yneJ acrR* double and single mutants to SDS treatment. Loss of *yneJ* increased sensitivity to SDS ($P = 6.4 \,\mathrm{e}^{-17}$), which was restored by overexpressing *yneJ* in trans. Conversely, *acrR* mutant exhibited resistance, implying that endurance to SDS is conferred through increased activity of AcrAB efflux pump. This finding is supported by reversing the resistant phenotype of *acrR* mutant by overexpressing *acrR* in trans (Fig. 4i). As with *acrR* mutant, *yneJ acrR* double mutants were also SDS-resistant, whereas overexpression of *acrR* in the corresponding double mutants displayed a similar SDS sensitivity to *yneJ* single mutant, suggesting that *yneJ* is likely to activate *acrAB* efflux pump by repressing AcrR regulation. This notion was supported in three ways. First, *yneJ* binding site upstream of *acrR* (i.e., intergenic region of *acrA-acrR*) that missed the *p*-value threshold from ChIP-seq peak calling was validated by ChIP-qPCR on native FLAG$_3$-tagged YneJ in M9- tryptone or putrescine, showing a 10- to 25-fold enrichment of YneJ occupancy (Fig. 4e) upstream of *acrR*.

Second, phylogenetic analysis on the orthologs of *acrR* across 54 closely related γ-proteobacterial species showed a low-degree ($P = 9.3 \,\mathrm{e}^{-2}$, Supplementary Fig. 4f) of sequence co-conservation (11%; 6 of 54) with no conserved motif detected within the *sad-yneJ* intergenic region. We inquired whether incubating the purified recombinant protein of YneJ-His$_6$ with AcrR-His$_6$ will enhance its binding to *sad* regulatory region. Thus, fluorescence polarization assay was performed with increasing concentrations of YneJ protein in the presence (0.5 and 1.0 μM) and absence of AcrR using the same 21 bp fluorescent-labeled oligonucleotides of the consensus motif in *sad-yneJ* intergenic region (Fig. 4b). Polarization increased considerably with YneJ protein when AcrR was added at both concentrations (0.5 μM data shown; Supplementary Fig. 4g, h) than YneJ without AcrR or with *leuO* negative control. This suggests that YneJ binds to *sad-yneJ* intergenic region efficiently in the presence of AcrR.

Third, consistent with RNA-seq (Supplementary Fig. 4b), qRT-PCR showed *acrAB* transcript level to be reduced by 2.1-to 2.5-fold, while *acrR* was enhanced by 2.7-fold upon *yneJ* deletion compared to wild-type in M9-tryptone. However, overexpression of *yneJ* in trans restored *acrAB* levels in *yneJ* mutant, while *acrR* level was near to wild-type (Fig. 4j). These findings led us to rename *yneJ* as *pggR* (for putrescine and Gln-Glu gene regulation) as our data supports a model (Fig. 4k) that under carbon limiting condition, *yneJ* regulates *yneI* operon, and activates *acrAB* pump by repressing *acrR* to efflux toxic metabolites from Gln-Glu or Puu pathways.

## Differential map uncovers GI rewiring to changing cultures.

Bacteria adapt to environmental distresses by regulating one or more TFs to modulate gene expression or interact with their target genes for transcription[3]. We thus examined static (RM, MM) maps, and identified the differences in GI score for each gene pair in RM vs. MM (i.e., $S_{MM}-S_{RM}$) for significance using an established scoring procedure[18]. A *P*-value significance assigned to a null distribution of score differences between RM and MM genetic maps resulted in a network of differential ($S_{DF}$) GIs (Fig. 5a). After applying a two-standard deviation cut-off ($-2.6 \leq$

$S_{DF} \geq 1.7$, $P \leq 0.05$) to the resulting network, we found 9208 significant differential aggravating or 7758 alleviating GIs (Fig. 5a and Supplementary Fig. 5a; Supplementary Data 9). Over half (59%; 10,063 of 16,966) of these differential GIs were too weak to detect in RM due to low epistasis scores, yet they exhibited a sizeable change in GI under a prototrophic condition (Fig. 5b).

Examination of all networks further indicated that interactions of TFs with known roles in metabolism, transport, stress response, and those with unknown function were more enriched in differential compared to static networks (Fig. 5c). The number of GIs for each gene in the differential network, and a gene's static GI profile altered by MM suggested a shift in gene function, and was also correlated with the sensitivity of the corresponding mutant strains in MM (Fig. 5d) or likely to be conditionally essential for growth in MM under different carbon sources (Supplementary Fig. 5b) from a recent genetic screen[46]. Consistent with this, GI profiles for one-fifth of TF genes (20%; 55 of 282) had high autocorrelation (> 0.25) in RM and MM, while the profiles for one-tenth (10%; 28 of 282) of TFs that had low autocorrelation (< 0.25) and were markedly disrupted in MM, were sensitive in MM from the phenotypic genetic screen[46] (Fig. 5e and Supplementary Fig. 5c; Supplementary Data 10). This includes deletion of a global TF, *lrp*, a leucine-responsive regulatory gene in metabolism[47], and *ydhB*, a LysR family regulator, whose role in TF that is less well understood, which showed low autocorrelation (Fig. 5e) with GI patterns between conditions, conferring growth defects in MM.

We next inferred differential GIs in conjunction with previously reported[48] *E. coli* functional modules (i.e., protein complexes and pathways) to understand how differential GIs occur in TF modules due to changing culture conditions. As noted previously[18], in contrast to static networks, differential GIs were not enriched within modules (Supplementary Fig. 5d), but rather enriched ($P = 2.2 \,\mathrm{e}^{-16}$) among gene pairs connecting two distinct modules (Supplementary Fig. 5e), suggesting that bacterial protein complexes are stable across conditions, and it is the GIs between modules that are reorganized under nutrient limitation. Using our permutation testing[49], differential GIs were transformed into a map of 33 modules enriched ($Z$-score $|\geq 1.65$; $P \leq 0.05$) for 48 inter-module interactions reconfigured in distinct growth conditions (Supplementary Data 11).

Besides recapitulating TFs that work in a coordinated manner, many differential GIs between modules have not been previously linked to TF mechanisms, allowing to infer testable hypotheses (Supplementary Fig. 5f). For instance, gene encoding for Crp and CytR (cytidine regulator) dimeric proteins, forming a nucleoprotein complex with opposing effects on transcription[50] showed distinct differential GI patterns. This includes known functional connections between the carbon and iron utilization regulators, *crp* and *fur* (ferric uptake regulator)[51], which was reflected as differential alleviating GIs, whereas *crp* that controls *lrp* expression[52], as well as repressors (*lrp*, *cytR*) that activate transcription manifested as differential aggravating GIs. Another module pair included the differential aggravating GI between *crp* and *fnr* (fumarate and nitrate reductase) global regulators, consistent with the requisite of both TFs for activation[53]. However, *crp* displayed differential alleviating GI with *melR* (melibiose-triggered transcription activator), supporting their joint binding to activate transcription in a co-dependent manner[54], while *fnr* that is physically linked with *melR* and *iclR* exhibited differential alleviating GIs with *cytR*, suggesting that global (*fnr*) and/or local (*melR*, *iclR*, *cytR*) TFs interplay share a likely regulator-target relationship.

To gain insights into GIs underlying static genetic changes between RM and MM, we used hierarchical clustering on 7369 significant ($P \leq 0.05$ and with a $|S\text{-score}| \geq 5.0|$ cut-off)

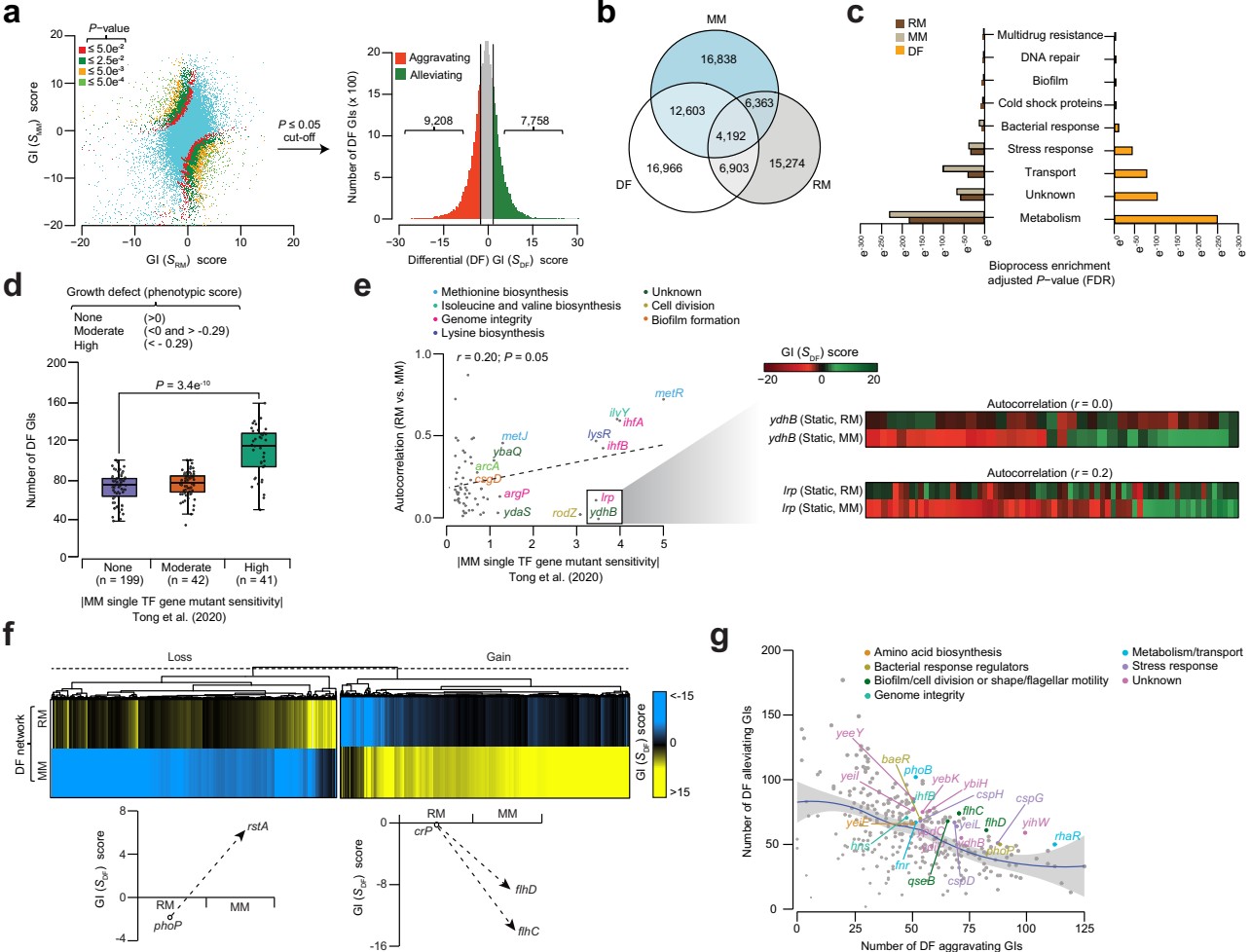

**Fig. 5 Differential networks reveal new TF gene functions. a** Scatterplot (left) of GI scores of RM vs. MM static networks at various *P*-value-based cut-offs (≤ 5.0 e$^{-2}$, 2.5 e$^{-2}$, 5.0 e$^{-3}$, 5.0 e$^{-4}$) after Z-score transformation. Histogram (right) of differential GI scores of the Z-score transformed *P*-value filtered at a significance level of 0.05 with tails representing significant epistatic interactions. **b** Venn showing the overlap of significant GIs in static and differential networks. **c** Enriched TF bioprocesses (*P*-value significance by hypergeometric test, and adjusted using the Benjamini-Hochberg FDR correction) in static and differential networks. **d**, **e** Distribution of differential GIs (**d**; from left to right, *n* = 199, 42, and 41) or correlation of GI profiles (**e**) for each query TF gene between RM and MM networks (i.e., Autocorrelation) is plotted against single TF gene mutant growth fitness sensitivity in MM from a recent phenotypic genetic screen[46]. Autocorrelation profiles for two representative query TF genes (*ydhB*, *lrp*) are shown (**e**); *P* = 3.4 e$^{-10}$ by two-sided Wilcoxon signed-rank test (**d**) and *P* = 0.05 by *t*-distribution for Pearson's correlation coefficient (**e**). Box plot (**d**) from left to right is shown with maximum (100, 102, and 159), minimum (39, 44, and 51), quartile 1 (Q1; 68, 72 and 93), Q2 (79, 80, and 118) and Q3 (82, 83, and 123) values. **f** Hierarchical clustering of GI patterns showing gene pairs with gain or loss of GI in RM or MM. Illustrative examples of two TF gene pair profiles are shown. **g** Differential GI hub TF genes in the indicated processes is plotted against the number of aggravating vs. alleviating differential GIs. Source data are provided as a Source Data file.

differential interactions that showed a marked GI shift in MM compared to RM (Fig. 5f). Of those, 3238 gene pairs were categorized as 'gain of interaction', while 4131 gene pairs as 'loss of interaction'. For example, the transcriptional regulator, *phoP* of the two-component system, that controls the transcription of an another two-component regulatory system[55], *rstA*, displayed a gain of interaction in MM compared to RM. Conversely, the global regulator, *crp*, that regulates the transcription of its target master motility complex, *flhDC*[56], showed loss of GI in MM.

**Differential map captures YdiP with CspA family members.** Since differential interaction hubs modulate a variety of cellular functions[18], and associated primarily with gene regulation[57], we probed the differential network and found one-fifth (22%; 66 of 302; Fig. 5g) of the TFs to be hubs (i.e., ≥ 100 GIs), and of those

47 were annotated to metabolism, transport and stress response, as well as to unknown function. We further explored one of the hubs identified in our differential network, *ydiP*, an uncharacterized HTH-type transcriptional regulator that showed strong GIs in MM (35%; 41 of 116) compared to RM (16%; 19 of 116; Fig. 6a), *ydiP-csp* gene pairs ranked in the top 1% in static MM network among all *ydiP* correlated GI profiles, and were not previously linked to the CspA family of cold shock protein (CSP) encoding genes (Fig. 6b).

CspA family encompasses 9 homologous proteins (CspA to CspI), with CspABEGI induced by cold shock[58], while other CSPs promote general stress adaptation responses[58]. Intriguingly, *ydiP*, showed prominent (*P* ≤ 0.05) differential aggravating and alleviating GIs with *cspACG* and *cspFDH*, respectively (Fig. 6b), suggesting that YdiP may have specialized functional roles with CspA family members that typically regulate differently[59].

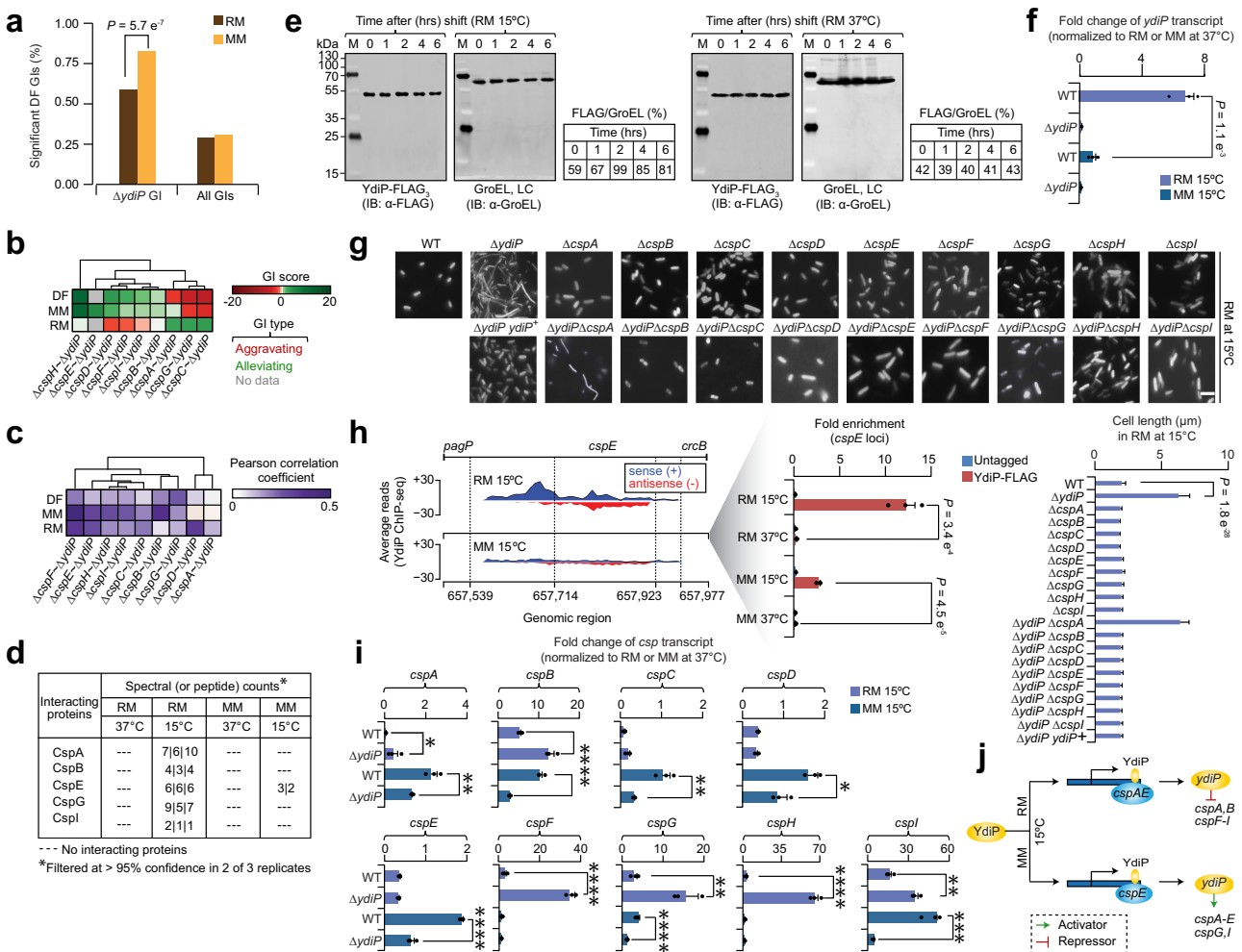

**Fig. 6 YdiP regulates cold shock proteins at low temperature. a** Significant ($P = 5.7\,e^{-7}$ by Student's two-sided $t$-test) differential GIs of *ydiP* with other TF gene mutants in RM and MM. **b, c** GI patterns (**b**) and correlation (**c**) profile of *ydiP* in static and differential networks with genes involved in cold shock adaptation. **d** YdiP interaction with indicated cold shock proteins in different growth and temperature conditions by affinity purification and mass spectrometry ($n = 3$ biologically independent experiments). **e** YdiP steady-state level in RM and at different temperatures over time. Band intensities normalized to *E. coli* Hsp60 loading control (LC; 1:15000 dilution). Immunoblot shown is representative of 2 independent biological experiments. Molecular masses (kDa) of marker proteins are indicated. **f** Transcript levels of *ydiP* measured in wild-type (WT) and *ydiP* mutant strain background in RM and MM at 15 °C are shown as fold change ($n = 3$ biologically independent experiments; $P = 1.1\,e^{-3}$ by Student's two-sided $t$-test) after normalizing to respective strains and growth conditions at 37 °C. **g** Representative cell morphology micrographs of *ydiP* and *csp* single and double mutants, as well as overexpression of *ydiP* (indicated with '+') in *ydiP* mutant strain in RM at 15 °C are shown along with cell length measurements ($n \geq 25$ cells over five biologically independent experiments; $P = 1.8\,e^{-28}$ by Student's two-sided $t$-test). Scale bar, 10 μm. **h** Mapped reads from YdiP ChIP-seq is plotted against the position of indicated genes in genomic region, including YdiP binding to its putative *cspE* target in RM and MM at 15 °C. Positive value indicate plus (or sense) strand, and negative value specify minus (or antisense) strand. Fold enrichment (shown as a zoom-in; $n = 3$ biologically independent experiments) of YdiP binding site to *cspE* using a chromosomal YdiP FLAG$_3$-tagged strain was compared to an untagged strain after normalizing to a negative control (i.e., primers designed away from YdiP binding site of *cspE* was amplified from YdiP- FLAG$_3$) in different growth and temperature conditions. Significance ($P = 3.4\,e^{-4}$ or $P = 4.5\,e^{-5}$) by Student's two-sided $t$-test between the YdiP FLAG$_3$-tagged strain grown in RM or MM at 15 °C and 37 °C. **i** Transcript levels of *csp* genes measured in WT and *ydiP* mutant in RM and MM at 15 °C are shown as fold change ($n = 3$ biologically independent experiments) after normalizing to respective strains and growth conditions at 37 °C. Significance (*$P \leq 0.05$, **$P \leq 0.01$, ***$P \leq 0.001$, ****$P \leq 0.0001$) by Student's two-sided $t$-test. **j** Model illustrating YdiP as an activator and repressor of various cold shock proteins in RM and MM growth conditions. Data (**f–i**) are presented as mean ± standard deviation from the indicated number of independent samples. Source data are provided as a Source Data file.

Consistent with this, *yjdC* displayed positively correlated genetic profiles with *csp* genes in the differential network (Fig. 6c), just as with the correlated phenotypic profiles observed between *ydiP* and *csp* single gene mutant strains in the phenomics screens (Supplementary Fig. 6a) performed under different drug treatments[60] and carbon environments[46]. Since functionally related genes are coupled physically[45], we affinity purified an endogenously C-terminally fused YdiP FLAG$_3$-tag strain grown under RM and MM at 15 °C and 37 °C, followed by mass

spectrometry, revealing YdiP binding with 5 (CspABEGI) of the 9 CSPs in RM, and with CspE in MM at 15 °C (Fig. 6d), but not under both cultures at 37 °C.

To understand the physiological role of YdiP at low temperature, the growth rate of *ydiP* and *csp* single mutants in RM was measured upon a temperature shift from 37 °C to 15 °C. In line with a previous report[61], *ydiP* and *csp* mutants showed no difference in growth rate at 37 °C when compared to wild-type. Conversely, *ydiP* mutant growth was impaired at 15 °C, just as

cold-inducible (*cspABEGI*) or stress response *csp* single mutants. However, this defect was exacerbated when both *ydiP* and *cspABEGI* were deleted, but not the *ydiP cspCDFH* double mutants which resembled one of the single mutants (Supplementary Fig. 6b), suggestive of functional dependency. A recent survey of bacterial ribosome profiling[21] in MOPS rich defined media at 37 °C and 6 hrs after shift to 10 °C showed that *ydiP* protein abundance was increased after adaptation to 10 °C relative to 37 °C to the same level as *cspCDE*, albeit to a lesser extent than *cspABFGHI* (Supplementary Fig. 6c). This agrees with our finding where YdiP steady-state level in RM was enhanced from 2 h after shifting to cold shock at 15 °C than YdiP in RM at 37 °C (Fig. 6e).

In contrast to *ydiP* mutant, the *ydiP* transcript level in wild-type strain grown under RM and MM, 4 hrs after shift to 15 °C, also exhibited a considerable increase in expression by 1.0- to 6.8-fold, respectively, compared to growing the respective strain under same conditions at 37 °C (Fig. 6f). Strikingly, contrary to 37 °C, *ydiP* mutant grown in RM or MM after 4 h at 15 °C formed long filamentous cells (~6.0 µm avg. cell length) relative to wild-type (~2.0 µm), while *ydiP* overexpression in *ydiP* mutant rescued the filamentous phenotype. In the case of strains lacking both *ydiP* and *csp*, the filamentous cells were similar to one of the single mutants in RM or MM at 15 °C, but not at 37 °C (Fig. 6g and Supplementary Fig. 6d, e; data not shown for MM at 37 °C). Together, these findings imply that YdiP target CSP-family proteins at low temperature.

We then sought to identify YdiP binding sites in RM and MM conditions 4 hrs after shift to 15 °C by performing ChIP-seq using a C-terminal chromosomal FLAG$_3$-tagged YdiP in an *E. coli* DY330 strain. As with YneJ, YdiP ChIP-seq peak-calling threshold was set to $P \le 1.0\,e^{-4}$, which corresponds to genes with start codons from 50 bp upstream and downstream of a binding site that was considered as potential YdiP targets. Remarkably, distinct patterns of ChIP-seq signals for YdiP were identified with 35 binding sites in RM, 8 in MM, and 9 in both culture conditions (Supplementary Data 12). While many of these sites were inside or in the intergenic regions of non-TFs, we found a binding site within *cspE* TF-encoding gene in RM (i.e., 35 bp from the transcription start site) but not under MM at 15 °C (Fig. 6h). While YdiP binding site that was detected with other *csp* genes in RM (i.e., *cspABDEHI*) or MM (*cspBDI*) at 15 °C were significant ($P \le 0.01$), the binding sites were far (i.e., −545 to +95 bp) from the predicted transcription start site of *csp* gene, except *cspA* (+79 bp); however, they all missed the stringent threshold set in the peak calling analysis. Hence, we examined YdiP occupancy at *cspE* loci by ChIP-qPCR, which enhanced on an average by 12.2-fold in RM or 2.7-fold in MM at 15 °C than in either growth conditions at 37 °C (Fig. 6h). Notably, though below threshold by ChIP-seq, ChIP-qPCR showed a 7.3-fold increase of YdiP occupancy at *cspA* loci in RM at 15 °C than at 37 °C (Supplementary Fig. 6f). These results suggest that CspAE loci are likely targets of YdiP at low temperature.

Since *ydiP* participates with CSPs in RM at low temperature, we examined whether *ydiP* contribute to the transcriptional regulation of CSPs under RM or MM at 37 °C and 4 h after shift to 15 °C by measuring *csp* transcripts in wild-type and *ydiP* mutant using qRT-PCR. Relative to 37 °C in respective growth conditions, *cspABFGHI* transcript levels were increased in *ydiP* mutant against the wild-type in RM at 15 °C, while *cspABCDEGI* transcripts in MM at 15 °C were reduced (Fig. 6i). This result suggests a model that *ydiP* (renamed as *csiP* for cold shock inducible protein) deletion influences the expression levels of *CSPs*, where *ydiP* acts as a regulator (Fig. 6j) by repressing and activating various *CSPs* under RM and MM, respectively, at low temperature.

**Epistasis connections among co-conserved TFs and paralogs.** As orthologous TFs across different phyla can reflect shared functionality, albeit in some cases with distinct roles[62], we investigated to what extent TFs with GIs in the static and differential networks are co-conserved in other bacteria. To address this, phylogenetic profiles were created for each TF in the network by retrieving orthologs across a total of 4409 species in 17 bacterial phyla from eggNOG database (Supplementary Data 13). While over one-third (37%; 16,151 of 43,365) of TFs with GIs in static networks were conserved, these associations in the assigned bioprocesses varied markedly across bacterial phyla (Supplementary Fig. 7a and Supplementary Data 14). These included almost one-tenth (7%; 164 of 2209) of TFs functioning in metabolism and bacterial response regulators connected by GIs in static networks exhibiting a strong propensity for co-conservation across all bacterial phyla, consistent with their broad functional significance. However, nearly two-thirds (62%, 102 of 164) of these conserved TFs were altered to changing culture conditions in the differential network. Conversely, TFs with GIs related to envelope biogenesis, cell division/shape, or flagellar motility were found in γ-Proteobacteria, Firmicutes, and Acidobacteria, but lacked orthologs in Thermotogae and were poorly co-conserved (2%; 1 of 62) among other bacterial phyla. This suggests that lack of co-conserved TFs may be due to horizontal gene transfer events, resulting in many ancestral TFs to be different among bacteria[62].

By limiting our analysis to the largest and most diverse bacterial phylum, γ-Proteobacteria (1729 genera), we found many enriched ($|Z\text{-score}| \ge 2$; $P \le 0.05$) co-conserved interprocess gene pairs connected by GIs to show notable differences in static and differential networks (Supplementary Data 15), such as the enrichment for conserved differential GIs between stress response and transport or translation, as well as between metabolism and bacterial motility (Supplementary Fig. 7a). As with the expectations[18], GIs with TF gene pairs involved in same bioprocess in static or differential networks tend to be conserved ($8.2\,e^{-9} < P > 1.4e^{-4}$) than those TF genes with GIs belonging to different bioprocesses (Supplementary Fig. 7b), suggesting that intraprocess GIs among co-conserved TF genes reflect functional associations that are imperative across closely- or distantly-related bacterial species.

We next probed GIs between paralogous TF gene pairs in static and differential networks to gain insights on how duplicated TFs endured to environmental changes across bacterial lineages. Aggravating GIs among 201 putative paralogous pairs, comprising 109 TF genes (Supplementary Data 16), appears to be more common in static MM and differential networks ($9.1\,e^{-12} < P > 1.0\,e^{-11}$) than random singleton gene pairs, representing that genetic redundancy between TF duplicates is modulated under stress conditions (Supplementary Fig. 7c). This assumption is consistent with the correlated GI patterns among TF paralog pairs in static and differential networks, which revealed that one-third of paralog pairs (34%; 68 of 201) had different GI profiles in changing conditions (Supplementary Fig. 7d). For instance, GI correlation profiles among the paralogs of OmpR family of 14 response regulators (*arcA, baeR, basR, cusR, cpxR, creB, hprR, kdpE, ompR, phoB, phoP, qseB, rstA, torR*) were altered in MM and differential networks. Distinct GI profiles observed among two homologous nucleoid-associated TF duplicate pairs, IHF (*ihfA, ihfB*) and HU (*hupA, hupB*) were altered in static RM and differential networks. These results suggest that epistasis among TF paralogs have a dynamic response to changing growth conditions.

## Discussion

To date, we have amassed GIs for over one-fourth (27%; 1184 of 4386) of the *E. coli* genome (Supplementary Fig. 7e), focusing on

various processes (i.e., cell envelope[19], protein synthesis[63], genome integrity[18]), yet TFs are underrepresented in the studies of GIs, impeding systems-level understanding. In this study, through TF epistatic screens we have provided a broader view of the overall robustness of TF systems, intricate condition-dependent epistatic connections, and crosstalk between TF processes under prototrophic or auxotrophic culture conditions. A notable observation is that in comparison to co-activators, co-repressed TF pairs appear to be contributing to the difference in epistatic connectivity. Local TFs are likely to drive gene expression due to changing growth conditions, which is expected to contribute to cell fitness and result in increasing numbers of aggravating GIs among local TFs, which was the case in this study and in yeast[34].

In addition to pathway crosstalk, we have utilized our genetic maps to reveal unanticipated dependencies and assign functions to unannotated TF genes not previously linked to cellular processes. For example, our static GI networks uncovered *yjdC* as a cell division regulator, as a repressor of *dicC* in a *dicA* mutant, and repress *dicA* and *dicB* when *yjdC* is exogenously supplemented. YjdC further found to bind to the operator region of *dicC*, enabling *dicA* expression to resume cell division. By mining static genetic maps, we also illuminated the function of *yneJ* to regulate *yneI* operon under carbon limiting conditions, and activate *acrAB* pump by repressing *acrR* to efflux toxic metabolites from the Gln-Glu or Puu pathway.

Alteration of TF GI profiles observed in different growth conditions is likely to modulate regulatory logic[34], suggesting that redundant regulation may be a general mechanism for rewiring in bacteria as reported for eukaryotes[34]. We have shown this for a non-essential gene of unknown function, *ydiP*, that acts as a regulator at low temperature by activating and repressing CSPs on RM and MM, respectively. The ability to mine static and differential networks to define genetic relationships extends from the quantitative fitness measurements of digenic mutants. To facilitate such explorations, we have created a portal (http://ecoli.med.utoronto.ca/eMap/TF) to examine the pair-wise interactions between TF processes in response to changing growth conditions.

Although the evolutionary basis of GIs among TF orthologs or paralogs in static and differential genetic networks remains abstruse, the ubiquity of TFs as a quintessential machinery in an increasing number of bacterial genome sequences has clearly acted as a key determinant of microbial evolution. This led to predicting condition-specific GI profiles and functions among TF orthologs or paralogs from *E. coli* to closely and distantly related bacterial phyla. For instance, among the co-conserved inter-process gene pairs, enrichment of GIs between TF orthologs of two-component signal transduction system (i.e., bacterial response regulators) and translation in static MM and differential networks across γ-Proteobacterial species suggests that adaptive changes of TF functionality in different culture conditions may have ascribed to condition-specific GIs. Nonetheless, given that regulatory circuits are often modified and expanded by bacteria[62], the generation of GI networks in *E. coli* suggests that functional vulnerabilities could possibly be exploited to understand how bacteria regulating their genomes via TFs to varying growth conditions contributes to phenotypic diversity across bacterial phyla.

In summary, we offer insights into epistatic ties among TFs towards understanding global transcription network, however more work is warranted to investigate: (1) how many TFs are interacting epistatically on differentially expressed TF genes in *E. coli*, and to what extent the GIs pertain to transcriptional outcome of a TF mutant; as well as (2) how GIs and promoter specificities of individual TFs have diverged from one another, and is there any overlap with their genome binding sites. Once protein-DNA and differential gene regulation in response to TF

perturbations becomes available for all TFs, which still remains incomplete[23], integrating these with GIs should provide unprecedented insight into the global gene regulatory circuits.

## Methods

**TF gene compilation.** A list of 304 non-redundant *E. coli* TF genes, including orphans of unknown gene function, was compiled based on published studies, annotations in EcoCyc, and RegulonDB database. These TF genes were classified into 41 minor bioprocesses based on peer reviewed literature curation, which were then collapsed into 13 major representative functional groupings and manually assigning each gene into more than one (i.e., fuzzy) bioprocess.

**Strains, plasmids, and growth conditions.** All bacterial strains, DNA oligonucleotides, and plasmids used are listed in Supplementary Data 17. In all experiments, strains were grown in LB (or RM), M9-MM with 0.1% glucose, and M9-Gln/Glu/tryptone/putrescine (MM with 0.5% Gln/Glu, 1% Bacto-tryptone, or 0.4% putrescine was used instead of 0.1% glucose) carbon source-limiting medium. For genetic screening, except *crp*, TF gene deletion mutant strains were obtained from the Keio F- recipient single gene knockout library marked with a kanamycin (Kan) resistance cassette[16]. Since *crp* mutant is partial diploid in *E. coli* Keio gene knockout collection, we constructed the corresponding gene mutant marked with kanamycin in house in BW25113 parental strain. Hfr C query TF single gene deletion donor mutants were created using λ-Red recombination system to replace the coding sequence of each non-essential gene with a chloramphenicol (Cm) selection marker, while hypomorphic alleles were generated by integrating selection cassette into 3'-UTR of essential genes to perturb transcript abundance, as described[15,19]. In-frame fusion of FLAG₃-tag to the natural C-terminus of the target protein was generated as previously[64]. LB broth or agar medium was supplemented with Kan (50 μg/ml), Cm (34 μg/ml), and/or ampicillin (100 μg/ml), as reported by us[15,18] and others[65] during the construction of double mutants.

Overexpression of target TF gene products was obtained from the ASKA library with Cm resistance marker[66], whereas overexpression of *acrR* in *yneJ acrR* double mutants was produced in house using a pACYC184 low copy plasmid with tetracycline-resistance marker. Briefly, genomic fragment containing endogenous promoter region and *acrR* gene were amplified by PCR from *E. coli* BW25113 strain using the forward and reverse primers (see Supplementary Data 17). The purified amplicon is then cloned with T4 DNA ligase into the pACYC184 cut site with HindIII restriction enzyme. The cloned product is transformed into the *yneJ acrR* double mutant competent cells, following standard transformation procedure, with positive transformed colonies selected on LB agar plates containing Kan (25 μg/ml), Cm (17 μg/ml), and tetracycline (5 μg/ml). In this experiment, we have lowered the Kan and Cm to half the dosage compared to eSGA genetic screens as we were able to detect resistant colonies to Kan-Cm even at levels below minimum inhibitory concentration. Besides, when handling such strains, we typically streak single colonies, and manually verify by PCR for the marker presence and gene deletions.

**Genetic screens, scoring, and other bioinformatic analyses.** TF genetic screens were performed as described[15], except that the last selection step was performed by replica-pinning viable double mutants (i.e., generated after 24 h of conjugation at 32 °C) onto MM containing both Kan and Cm antibiotics, and incubated for an additional 24 h at 32 °C. Each Hfr donor query TF gene mutant was screened twice against an arrayed F- recipient single gene deletion mutants in quadruplicate, generating eight replicate pairs per donor for statistical reproducibility. Colony image analysis, processing of epistatic data, and static and differential GI score generation using multiplicative, or machine learning-based Gaussian models were calculated as described[18,20]. Most notably, our assignment of GI score based on colony size measurement for each digenic mutant using either one of the scoring models were not ranked based on *P*-value. Instead, it is to help interpret the significant change in the fitness of double mutants compared to random chance.

Bioprocess enrichment analysis was performed on the static and differential GI networks as previously[18] by considering 2 or more TF genes from each minor and major functional grouping. Enrichment of intra- or inter-process GIs was computed based on a hypergeometric cumulative probability distribution using the ensuing parameters: $k =$ number of significant GIs between bioprocesses, $n =$ number of all possible GIs between bioprocesses, M = number of significant GIs observed in the network, and N = number of possible GIs in the network. Module crosstalk analysis in differential network was performed by mapping significant GIs to a set of *E. coli* functional module predictions[48] using a permutation test as previously described[18]. Each module membership of the target genes was randomly reassigned for 1,000 interactions, and the resulting inter-module GI distribution from randomized network(s) was transformed to a Z-score. Only inter-module pairs with Z-score $|\geq 1.65$ (with $P \leq 0.05$) was considered as significant.

To assess the conservation of GIs among TF genes in static and differential networks, a phylogenetic profile for each TF gene pair was constructed by retrieving orthologous groups across 4409 species in 17 bacterial phyla from EggNOG database (5.0 ver., downloaded Nov 2020). Using a statistical framework as previously[18], we considered a GI pair to be evolutionarily conserved only if they

were detected in all bacterial phyla. The orthologous TF gene pairs for each phylum were then annotated to the same or different bioprocesses. Caution is needed while interpreting co-conserved gene pair comprising an epistatic interaction as mapping orthologous genes is not sufficient to guarantee the existence of GI in other species and it warrants detailed experimental investigation. Next, *E. coli* TF paralogs in the static and differential networks were identified by BLASTP sequence alignment search, satisfying ≥ 50% in alignment length with a sequence similarity score of *E*-value ≤ 5 e$^{-2}$, and encoding a 30% amino acid sequence identity.

To identify the putative YneJ binding site for *sad* regulation through conservation analysis, we subjected *sad-glsB-yneG* and *yneJ* DNA sequences against γ-proteobacterial species of Gram-negative bacteria to nucleotide BLAST using default settings, resulting in 52 hits with ≥75% sequence identity. The BLAST Tree View Widget that clusters sequences according to distances from query sequence was then used to construct a phylogenetic tree with *sad-glsB-yneG-yneJ* sequences, which exhibited conservation in 23 species of the Proteobacterial phylum. Among those, we found a consensus binding site in *sad-glsB-yneG-yneJ* intergenic region with sequences conserved in 8 Proteobacterial species using Clustal Omega multiple sequence alignment. Similar analysis was performed for *acrR* across γ-proteobacterial species.

**Growth curve and phenotypic assays**. To validate GIs or determine growth fitness for a set of double and single mutant strains by liquid growth assays, we inoculated overnight cultures of wild-type and TF gene deletion mutants into 96-well microtiter plates containing 100 μl of LB and/or MM at varying temperatures (15 °C, 32 °C, 37 °C). The plates were incubated with shaking for over 24 h, and absorbance of the culture was measured at an optical density (600 nm) using an automated synergy plate reader. To examine cell morphology, wild-type and mutant strain cultures were grown to an OD$_{600}$ ~0.5 log-phase in LB and/or MM at 15 °C, 32 °C, or 37 °C. The pelleted cells were re-suspended in cold phosphate buffer saline (PBS), and about 50 μl of suspended cells were spotted onto a poly-L-Lysine coated glass slide for imaging in a Zeiss AxioVert.A1 inverted epi-fluorescence microscope with cell length measured using ImageJ plugin.

**Bacterial two-hybrid**. The *E. coli* BTH101 strain was grown overnight at 37 °C in RM supplemented with ampicillin and Kan for vector (pUT18C, pKT25) propagation. Cultures were diluted in fresh LB broth and grown at 37 °C until OD$_{600}$ is between 0.4 and 0.5, followed by spotting on M63 agar plate (maltose as a carbon source) containing ampicillin, Kan, IPTG (isopropylthio-β-galactoside) to induce gene expression, and 40 μg/ml of chromogenic substrate 5-Bromo-4-chloro-3-indolyl-β-D-galactopyranoside (X-gal). Plates were incubated at 32 °C for 16–24 h, followed by monitoring color change. If two proteins physically interact, reconstitution of adenylate cyclase (CyaA) enzyme, leads to the release of a blue dye from the chromogenic substrate X-Gal as an indicator of positive interaction.

**Affinity purification-mass spectrometry**. FLAG$_3$-tagged C-terminus strains with a Kan marker integrated by homologous recombination into the *E. coli* chromosome were grown to mid-log phase in 100 ml of LB, MM, M9-putrescine or tryptone medium at 15 °C, 32 °C, or 37 °C. The pellets isolated from the harvested cells after centrifugation at 4,000 *xg* were resuspended in RIPA buffer (150 mM NaCl, 50 mM Tris-HCl (pH 7.5), 0.1% SDS, 1% Na deoxycholate, 1% NP-40, 1 mM EDTA), followed by sonication for 1 min with 30 sec cooling time in between for 8 cycles. The lysates were centrifuged at 12,000 *x g* for 20 min at 4 °C to remove debris, and purified by MACS® microbead method (Miltenyi) using magnetic columns, followed by washing the column three times using RIPA buffer containing detergent, and two washes with deterrent-free buffer. The Speedvac dried samples were trypsin digested and subjected to an Orbitrap Elite mass spectrometer, and the mass spectra were mapped to the reference W3110 *E. coli* protein sequences using SEQUEST (ver. 27 - rev.9) and STATQUEST (ver. 10) algorithms as previously[64].

**Co-immunoprecipitation and immunoblotting**. High-copy number plasmid pCA24N with a Cm resistance cassette for selection, and IPTG-inducible promoter P$_{T5-lac}$ controlling the expression of an N-terminal histidine (His$_6$)-tagged protein from ASKA overexpression library[66] was isolated using a Qiagen plasmid Miniprep kit. The resulting plasmid DNA transformed into an endogenously FLAG$_3$-tagged strain marked with Kan that is under the control of its native promoter is expressed in DY330 strain. The positive transformants on Kan-Cm selection were grown at 32 °C in 50 ml LB medium with both antibiotics until the cells reached an OD$_{600}$ of 0.6, at which point the cells were induced with 1 mM IPTG and grown for an additional 2 h. Harvested cells centrifuged at 4000 *x g* for 15 min was lysed using sonication, followed by centrifugation at 12,000 *x g* for 20 min to collect supernatant for immunoprecipitation, which was performed using the MACS® microbead method. Eluted samples were analyzed by immunoblotting via resolving them in 10% SDS-PAGE gel, transferring them onto a nitrocellulose membrane, probing with a His$_6$-tag monoclonal antibody, and visualizing by chemiluminescence.

**ChIP-seq and ChIP-qPCR**. FLAG$_3$-tagged strains were grown in LB, MM, M9-putrescine/tryptone medium at 15 °C, 32 °C, or 37 °C to an OD$_{600}$ of 0.3-0.5.

Harvested cultures were crosslinked, sonicated, and chromatin was immunoprecipitated, with following modifications. Monoclonal anti-FLAG antibody (1:5000 dilution) was used for immunoprecipitation in ChIP-seq or ChIP-qPCR, while input and mock samples for ChIP-seq were immunoprecipitated with rabbit (1:80 dilution) and mouse (1:135 dilution) immunoglobulin G antibody, respectively. ChIP-seq libraries were created and sequenced on a single lane of an Illumina Hi-Seq 2500 (paired-end reads, 2 × 125 bp) at the Toronto Hospital for Sick Children sequencing facility. The raw ChIP sequence reads were mapped to the *E. coli* W3110 chromosome using bowtie2 (ver. 2.4.0) software. After removing duplicates, the uniquely mapped reads sorted by samtools were subjected to Bioconductor MOSAiCS (MOdel-based one and two Sample Analysis and Inference for ChIP-seq; ver. 2.9.0) package for peak calling. Artifactual regions were excluded if peaks called in the ChIP-seq were present in input and/or mock samples.

In the case of ChIP-qPCR, FLAG$_3$-tagged strains were grown as in ChIP-seq, with oligonucleotides reported in Supplementary Data 17 are designed to amplify target binding and negative control regions. The latter was created by amplifying the primers designed away from YneJ binding site of *sad* or *acrR* from YneJ-FLAG$_3$ or YdiP binding site of *cspA* or *cspE* from YdiP-FLAG$_3$. The resulting ΔCq values from negative control was subsequently normalized to target ΔCq values from the actual ChIP-qPCR experiments.

**RNA isolation, qRT-PCR, and RNA-seq**. Wild-type and mutant or overexpressed strains with Kan or Cm selectable marker were grown in LB, MM, M9-Gln/Glu/putrescine/tryptone medium at 15 °C, 32 °C or 37 °C to an OD$_{600}$ of 0.4–0.5. Overexpression was performed by inducing the strains with 1 mM IPTG, and incubated at 32 °C for an additional 2 h. In the case of cold shock experiments, overexpressed culture was shifted to 15 °C for 4 h. Total RNA was isolated using Qiagen RNA extraction kit, treated with DNAse (Qiagen), and reverse transcribed using iScript reverse transcriptase (BioRad), following manufacturer's protocol. The resulting cDNA samples were diluted and used as template for qRT-PCR, which was performed in a Roche real-time LightCycler 96 PCR machine. Oligonucleotides used to amplify TF and housekeeping (*hcaT*) gene are described in Supplementary Data 17. Relative expression values were calculated using a standard $2^{-\Delta\Delta Cq}$ method, where target ΔCq values were normalized to housekeeping ΔCq values, and the changes in transcript levels were expressed in fold change (i.e., mutant over wild-type).

The RNA-seq experiment was conducted using the DNAse-treated RNA isolated from *yneJ* deletion mutant and wild-type BW25113 parental strain grown in M9-tryptone media as described above. The strand-specific DNA libraries were constructed after depleting the abundant ribosomal RNA (rRNA) with Illumina's stranded total RNA prep, ligation with Ribo-Zero plus kit. Sequencing was performed on an Illumina NovaSeq 6000 instrument (paired-end reads, 2 × 100 bp) at the Toronto Hospital for Sick Children sequencing facility. Using Salmon algorithm (ver. 1.4.0), the sequence reads were mapped to the *E. coli* W3110 reference genome, and subsequently performed an alignment-free based quantification to determine the transcript abundance. This was calculated based on the number of reads mapped to a gene divided by the gene length and normalized to the mean number of reads per kilobase of transcript per million sequencing reads. Since we were interested on the transcript level changes for genes in Gln-Glu and Puu pathways, as well as for *acrR-acrAB* efflux pump, we conducted gene expression analysis by computing the log$_2$ fold change of *yneJ* mutant vs. wild-type, and reported changes in gene expression for those genes without setting a 2-fold statistical significance to consider as differentially regulated.

**Protein purification, and fluorescent polarization assay**. The plasmids carrying an N-terminal His$_6$-tagged YneJ or AcrR TF protein was grown to mid-logarithmic phase in 1 L LB broth, and the tagged protein was expressed with 1 mM IPTG induction for 2–4 h. Cells were harvested by centrifugation at 4000 *x g* for 15 min, and lysed by passing it through sonication, and centrifuged additionally at 12,000 *x g* for 20 min to pellet cell debris. The resulting supernatant was added with an equal volume of equilibration buffer (i.e., PBS with 6 M guanidine-HCl and 10 mM imidazole; pH 7.4), and was subsequently loaded onto a 10 mL HisPur Ni-NTA resin column (Thermo Scientific). After collecting flow-through, the column was washed with 20 mL of wash buffer (i.e., PBS with 6 M guanidine-HCl and 25 mM imidazole; pH 7.4), and the eluate was eluted from the column with 20 mL of elution buffer (PBS with 6 M guanidine-HCl and 250 mM imidazole; pH 7.4). The recombinant protein was separated by size-exclusion chromatography on a Sephadex G-25 gel filtration column (Roche) to remove additional contaminants. The purified protein was assessed by SDS-PAGE, and/or immunoblotted using an anti-His antibody (1:1000 dilution).

The YneJ association with its cognate Sad binding site predicted from conservation analysis was determined by fluorescent polarization DNA-binding assay. Briefly, 3′ end of a DNA strand with 21 bp oligonucleotides (synthesized from Integrated DNA Technologies) was labeled by a fluorescent probe, 6-carboxyfluorescein, and annealed to 5' end of the strand with an unlabeled complementary 21 bp oligonucleotides at a 1:10 ratio. The resulting fluorescence-labeled double-stranded DNA fragment of the consensus motif in the *sad-yneJ* intergenic region with increasing concentrations of the recombinant YneJ protein, with or without AcrR recombinant protein or Gln, Glu and putrescine metabolites at various dilutions was added to a 100 μl reaction mixture in 96-well black plates

containing the binding buffer (phosphate-buffered saline (pH 6.5), 0.5 mM EDTA, 7 mM MgCl$_2$, 100 mM KCl, 5 mM phosphate). The plates were incubated at 50 °C in the presence of Herring sperm DNA to inhibit non-specific binding. The fluorescence-labeled DNA was detected using a Synergy multimode fluorescence microplate reader. Similar procedure was followed up with 28 bp oligonucleotides designed within the promoter region of *leuO* gene that is supposedly not to bind to *yneJ* or *ydiP* based on ChIP-seq data served as a negative control.

**Reporting Summary**. Further information on research design is available in the Nature Research Reporting Summary linked to this article.

## Data availability

Data that support the findings of this study is available at our resource website ([http://ecoli.med.utoronto.ca/eMap/TF](http://ecoli.med.utoronto.ca/eMap/TF)). ChIP-seq data have been deposited in NCBI under the BioProject accession number PRJNA771186, and are publicly available. Paralogous transcription factor genes passing the set criteria were assigned to bacterial non-supervised orthologous groups (BactNOGs) predicted by EggNOG database. Gene sequences from prokaryotic species were retrieved from NCBI non-redundant protein database to perform conservation and phylogenetic analyses. Source data are provided with this paper.

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

## Acknowledgements

We thank W. Houry for providing GroEL (HSP60) antibody, and express gratitude to the former members of Babu laboratory for their technical assistance. A.G., M.J., and M.T.M. are supported by the Canadian Institutes of Health Research (CIHR) and/or the Saskatchewan Health Research Foundation postdoctoral fellowships. M.B. is a Chancellors Research Chair in Network Biology. This work was supported by grants from the Natural Sciences and Engineering Research Council of Canada (DG-20234), and the Canada Foundation for Innovation to M.B.

## Author contributions

M.B. conceived and supervised the project. A.G., A.H., Z.I., M.T.M., M.Z., H.A. S.K., and M.J. contributed to the experiments and discussion of the results. M.R., S.P., and Q.Z. performed and coordinated the data analysis with guidance from A.G., K.A., and M.B. S.P. developed the web portal. A.G., K.A., and M.B. wrote the manuscript with input from M.J. and others. All authors contributed, read, and approved the manuscript.

## Competing interests

The authors declare no competing interests
