## [Peer Review File · Nature Communications]

Reviewers' Comments:

Reviewer #1:

Remarks to the Author:

To reveal transcription networks is one of the advanced research for understanding biological systems. However, the network is very complex, because transcription factors form a hierarchical network. In this study, the authors constructed double gene knockout strains of transcription factors to identify the genetic network of transcription factors in the best-understood organism, *E. coli*. The series of knockout strains were subjected to observe the colony growth on rich and poor medium, and a list of genetic interactions was provided by different growth between single gene knockout strains and double gene knockout strains. Based on the list, three function unknown transcription factors were studied in detail, and the physiological function and the newly transcription networks are proposed.

The research topic is very important for understanding biological system and the approach of using double gene knockout to identify the transcription factor genetic networks is interested and unique. The list of transcription factor genetic interactions based on the growth differences is valuable and useful for identification of genetic networks. However, in the transcription regulation analysis specifically addressed in the second half part of this manuscript, the proof of direct regulation and physiological role is poor. In addition, just arranging the results is not enough to consider the function and physiological significance of test transcription factors. Most experiments have been observed with only one point, and the effects of the concentration and the degree have not been verified. For this reviewer, the regulation of test transcription factor to a regulon of transcription factor of another genetic interaction pair seems indirect, thus, that makes the data of genetic interaction less credible. This reviewer agrees that the genetic interaction data indicate the functional interactions between test transcription factors, but disagrees for the direct regulation interactions.

To substantiate author's claim, evidence of direct regulation and plausible physiological data is needed. There may be an idea of splitting this manuscript into several manuscripts, such as genetic interaction part and individual transcription factors, respectively.

Major comments

1. To evaluate the data of genetic interactions between transcription factors, GI scores were calculated. Based on the GI scores, the authors picked up the several case and illustrated in Fig. 2, such as *crp-fis*, *hns-stpA*, and *baeR-cpxR-ppfF*. Are these GI scores significant compared to the others? For example, in the case of *crp-fis* in RM, P-value of MP score is less than 4000th, and P-value of GP score is less than 5000th. Does this mean that more than 4000 pairs of transcription factors exhibit genetic interaction? Why did the authors select these pairs, why didn't the authors explain the more plausible GI score? Can the authors objectively assess the GI score and the genetic interaction between transcription factors? Throughout the manuscript, don't be arbitrary, the authors need to objectively explain why they picked up the GI with the GI scores compared with other GI scores including *yjdC-dicA*, *yneJ-acrR*, and *ydiP-csps*.

2. The transcriptional interaction between *dicA* and *yjdC* is unclear. 2-1. The authors claim that *yjdC* is regulated by *DicA*, and regulates *dicA*, *dicC*, and *dicB*. However, only transcriptional levels of the test gene were measured, and indirect effects could not be ruled out. To understand the transcriptional network, the proof of direct regulation is mandatory. 2-2. The expression levels of *dicB* and *dicC* were increased in *dicA yjdC* double knockout strain. Thus, the formation of filamentation in *dicA* mutant was occurred by *dicB* and *dicC* expression, *dicA yjdC* double knockout strain should form the filamentation, but not. These data indicate that the reason of filamentation is not only the expression level of *dic* genes, and it suggest that there are other *yjdC* target genes involved in formation of filamentation, or *YjdC* senses the environmental change which occurred by *dicA* regulon. 2-3. This reviewer does not think that there is no functional interaction between *dicA* and *yjdC*.

3. The transcriptional interaction between *yneJ* and *acrR* is unclear. 3-1. Why did the *yneJ acrR* double mutant show growth defects under rich medium and minimal medium in GI screening? There is no clear explanation based on the features of *yneJ* and *acrR*. 3-2. The authors claim that *YneJ* regulates *sad-glsB-yneG* operon and *yneJ* mutant showed growth defect under *Gln*, *Glu*, and *Putr* as a sole carbon source (Fig.S3a). For *Glu* and *Putr* catabolism, there are isozymes shown in

Fig. 4D. Are these isozymes also under regulation of YneJ? 3-3. The expression levels of yneJ, sad, glsB, and yneG were increased in M9+Gln/Glu/Putr/Try compared to rich medium (Fig. 4E). Why compared it to rich medium? yneJ, sad, glsB, and yneG genes can be increased in minimal medium. To show the importance of metabolites, the control medium must be M9 medium containing other carbon source (glucose). If say further, similar induction levels were observed in M9+Tryptone medium, suggesting that induction does not appear to be specific by Gln/Glu/Putr. yneI and yneH had been renamed to sad and glsB, respectively. 3-4. In the ChIP-seq screening, the YneJ binding sites were altered between M9-putrescine and M9-tryptone medium. Under these two conditions, the target sequences were completely different (Table S8). What is the mechanism by which the target sequence of a transcription factor is altered in different media? The most suspicious is that the promoter region of sad-glsB-yneG operon could not be detected. The authors state that they set the cut-off and removed non-specific peaks in ChIP-seq (P16 L9). Along this line, YneJ binding level on sad detected by qPCR is also non-specific. 3-5. YneJ recognition sequence in sad promoter is suspicious. In Fig. S3c, surrounding sequence is also conserved, and the highlighting sequence is not specific. The resolution of ChIP-seq or ChIP-qPCR is hard to determine the consensus sequence. Is there any consensus sequence among the YneJ targets in Table S8? 3-6. In gel shift assay (Fig. 4g), non-specific binding has not been ruled out. Another DNA sequence should be tested for negative control. Effector test should be performed in different concentration manner. Other metabolites (at least Glu and Putr) should be tested to show the specific effect of Gln. 3-7. To test the yneJ regulation, the effect of Gln addition should be tested. Why sad-glsB-yneG did not induce in rich medium (Fig.4e)? Gln is included in rich medium. 3-8. In Fig. 4f. The control should be set as YneJ-FLAG under M9+glucose. The untagged sample is just an experimental control. Also, there is no control for input (background) (Fig. S3f). 3-9. What happens to AcrR that forms complex with YneJ? If the DNA binding affinity changes, it can be tested in vitro, such as by using gel shift assay. 3-10. acrR mutant did not show the growth difference against SDS stress in Fig. 4i. This result suggests that acrR did not involve in SDS resistance? AcrR is repressor of acrAB operon. If YneJ acts as a repressor for AcrR, SDS sensitivity reduces in yneJ mutant, but the result showed opposite effect. 3-11. Fig. S3b. Is AcrR also conserved between YneJ conserved strains? 3-12. The author's idea that yneJ responses to compounds in putrescine degradation pathway was derived from the function of sad-glsB-yneG operon. The gene names of sad and glsB must be used.

4. The transcriptional interaction between ydiP and CSPs is unclear. 4-1. Why did ydiP GIs show opposite effect between cspACG and cspFDH? This effects were inconsistent with growth test in Fig S5b (P23L18). Why? Furthermore, in Fig. 6g and Fig. S5d, the cell form of ydiP was restored by causing double mutants with either csps gene. This effect does not match the GIs effect. 4-2. P24L1. The authors claim that YdiP expression level was enhanced from 2hrs after shifting to cold shock, but in Fig 6e, YdiP expression levels were similar at 0hr and 2hrs. 4-3. P24L3. The authors claim that ydiP transcript level was enhanced from 4hrs after shifting to cold shock, but YdiP expression levels were similar at 0hr and 4hrs. 4-4. P24L18. Why the authors set the cut-off for 50 bp upstream and downstream from start codon in YdiP ChIP-seq screening? In the case of YneJ, they set 200 bp from start codon. In Table S12, What is the mechanism by which the target sequence of a transcription factor is altered in different temperature? 4-5. The interaction between YdiP and cspE was shown, however, what about other csp genes? At least, it is not enough to prove that YdiP acts as a dual regulator for CSPs. Including cspE, functional interactions between ydiP and csps can be indirect effects.

Minor comments

Identification of genetic interaction was performed under rich medium and minimal medium and many interactions showed different growth between media. However, little explanation is given based on the function of transcription factor. Thus, the credibility of the data is reduced.

P11L10, There is a publication showing that ygfI gene is involved in oxidative stress resistance. DOI: 10.1038/ncomms1146

P3L14, References 10 and 12 are not in vivo ChIP screening, in vitro genomic SELEX screening.

To rename the gene name should be done carefully. The transcriptional role of ygfI and yneJ is still

unknown.

Reviewer #2:

Remarks to the Author:

Manuscript ID; NCOMMS-21-41045-T

Title; Auxotrophic and Prototrophic Conditional Genetic Networks Reveal the Functional Rewiring of Transcription Factors in *Escherichia coli*

Authors; Alla Gagarinova, Matineh Rahmatbakhsh, Ali Hosseinnia, Zoe Istace, Sadhna Phanse, Mohamed Moutaoufik, Mara Zilocchi, Qingzhou Zhang, Hiroyuki Aoki, Sunyoung Kim, Khaled Aly, and Mohan Babu

This paper described the genetic interaction of 302 predicted transcription factors, including experimentally verified and unknown functions, in *E. coli* K-12 by systematic construction of double deletion strains by conjugation and analyzed on rich medium and minimal medium, or differential analysis when switching from rich medium to minimal.

This group was an original member of the group that established the method for analysis of genetic interaction in *E. coli* in 2008 (Butland et al.), and have been continuously performing genetic interaction analysis in *E. coli* since then, by conjugation method to construct double knockout strains.

The authors performed statistical analysis appropriately together with the structural and functional information of the transcription factors. In addition, some of the hypotheses obtained from the genetic interaction analysis were experimentally verified for a group of transcription factors with entirely unknown functions, showing that functional prediction is possible. The analysis performed is a large-scale analysis, and their analysis was carefully designed and performed using reliable methods.

This paper showed a new way to elucidate the dynamic rewiring of the transcriptional network and should be beneficial for the readers after addressing some of the small concerns listed below.

1) In the last paragraph of the abstract, the authors mention pathogenesis, but there does not seem to be any particular analysis to pathogenesis in the analysis. Moreover, it seems too abrupt. Is it really necessary?

2) In the results, the authors described the 304 transcription factors targets, and out of these, 278 were converted to Hfr by the lambda RED recombination method. On the other hand, Supplemental Table 2 showed a list of interaction pairs, and Gene 1 and Gene 2 listed 301 and 300 genes, respectively. The recipient strains were clearly from the Keio collection, but the donor side strains were not clear. Therefore, preparing the table listed Hfr strains is clearer, including conditional deletion strains used in the experiment.

3) Page 5, line 10, is there any reference to the statement, "mazE-ala-tRNA has a significant effect on growth"? If yes, it should be cited.

4) Page 5, line 14, In the case of transferring from RM to MM, the effect of changing the culture medium may differ significantly depending on the deleted gene. In particular, those related to trace elements are likely to be greatly affected by the change of the medium. What measures have the authors taken in this regard? Couldn't it have been done with MM from the beginning, including the conjugation step?

5) Page 8, line19, crp deletion strains could not be purified in the Keio collection (Yamamoto et al.). How did the authors use crp deletion strains for this analysis? It seems highly likely that the strain is a partial diploid. Did the authors purify them?

6) Page 18, line 16, "other yneJ interacts physically with acrR, with which it showed an aggravating GI." Genetic interactions between genes, whose products form protein complexes, are thought to be alleviating, but this combination showed an aggravating interaction. However, this combination shows an aggravating interaction experimentally, so what is the reason for suspecting a physical interaction to be confirmed? Does it mean that YneJ forms a complex with AcrR but

ultimately regulates the function of AcrR as a TF, forms a complex is not essential for the AcrR TF function? Is this what the authors considered?

7) Page 31, line 21, very high concentrations of Kan (50µg/mL) and Cm (34µg/mL) are used for the single copy drug resistance gene in the genome. However, in the next page 32, line 7, reasonably low concentrations of Kan(25µg/mL) and Cm(17µg/mL) for the low copy plasmid are used for the pACYC. High concentration of antibiotics might affect the frequency of partial diploid appearance, have the authors checked this concerning issue?

8) Figure 1, a) $j\Delta::Cm$ in the right figure seems to be a mistake for $j\Delta::Km$.

9) Supplementary Fig. 1, c) left Aggravating graph, why is the second bar graph shown with green dots? Is it a mistake for red?

10) Supplementary Fig. 4, b) Is this a different form of the analysis by Tong et al. as a graph? Please check if this is an appropriate citation.

11) Finally, regarding the gene ID of E. coli, the b number is one ID, but the ECK ID should be included since it has been adopted by the international E. coli community. Cooperatively developed annotation snapshot--2005. Nucleic Acids Res 34, 1-9 (2006). PubMed: 16397293.

Reviewer #3:

Remarks to the Author:

In this work, Gagarinova et al. use the high-throughput E. coli synthetic genetic array strategy to explore the large-scale rewiring involving 304 E. coli regulators between auxotrophic and prototrophic conditions as defined by nutrient rich medium and minimal medium + glucose, respectively.

The work is interesting as provides new insights in how the regulatory machinery globally adapts to two different and opposite conditions. The set of epistatic interactions identified in this work conforms a map that provides interesting data to improve our understanding of the organization of global gene circuits. It also provide new putative functions for uncharacterized regulators and analyzes the co-conservation of genes involved in epistatic interactions.

The manuscript is well written, clear, despite a bit longer than usual but maybe justified given the amount of work involved. Data generated in the work is reported as supplementary material and in a web portal developed by the authors (<http://ecoli.med.utoronto.ca/eMap/TF>).

Major suggestion:

- The authors investigate to what extent their epistatic interactions among TF are "generalized" in other bacteria. To answer this they use phylogenetic profiles constructed using COGs (clusters of orthologous groups). However, the co-conservation, via orthology, of the source and target genes comprising an epistatic interaction is not sufficient to guarantee the existence of an epistatic interaction. Authors must further clarify this point to the reader.

Minor suggestion:

- Supplementary figure 2, panel d: There is a typo in "Bacterial response", it must be "Bacterial response"

Response to Reviewers

We thank the reviewers for their insightful comments and constructive suggestions to bring more clarity to the manuscript. As detailed below, changes addressing the concerns raised include: (1) *yjdC* regulation by *dicA*, which is now supported by the inclusion of *yjdC* binding to cell division regulatory (*dicAC*) and inhibition (*dicB*) genes by chromatin immunoprecipitation quantitative PCR (ChIP-qPCR), (2) specificity of glutamine (Gln) effect is reinforced by testing other metabolites such as glutamate (Glu) and putrescine, and YneJ binding to *sad-yneJ* regulatory region is now shown to be enhanced with increased concentrations of YneJ in the presence of AcrR as evidenced by fluorescence polarization assay; as well as (3) quantitative real-time PCR (qRT-PCR) and ChIP-qPCR datasets normalized to appropriate controls. In addition to our point-by-point responses below, we have revised the main text and figures to address the reviewer's concerns, aiming for clarity with sufficient detail, and ensuring that the conclusions in the main text reflect the data presented in the figures. We feel that the revised manuscript presents our work more clearly, and will make it more broadly accessible to the readers.

Reviewer #1:

Comment # 1: To reveal transcription networks is one of the advanced research for understanding biological systems. However, the network is very complex, because transcription factors form a hierarchical network. In this study, the authors constructed double gene knockout strains of transcription factors to identify the genetic network of transcription factors in the best-understood organism, *E.coli*. The series of knockout strains were subjected to observe the colony growth on rich and poor medium, and a list of genetic interactions was provided by different growth between single gene knockout strains and double gene knockout strains. Based on the list, three function unknown transcription factors were studied in detail, and the physiological function and the newly transcription networks are proposed. The research topic is very important for understanding biological system and the approach of using double gene knockout to identify the transcription factor genetic networks is interested and unique. The list of transcription factor genetic interactions based on the growth differences is valuable and useful for identification of genetic networks.

Response: We thank the reviewer for the positive feedback and their valued comment on the importance of our study.

Comment # 2: In the transcription regulation analysis specifically addressed in the second half part of this manuscript, the proof of direct regulation and physiological role is poor. In addition, just arranging the results is not enough to consider the function and physiological significance of test transcription factors. Most experiments have been observed with only one point, and the effects of the concentration and the degree have not been verified. For this reviewer, the regulation of test transcription factor to a regulon of transcription factor of another genetic interaction pair seems indirect, thus, that makes the data of genetic interaction less credible. This reviewer agrees that the genetic interaction data indicate the functional interactions between test transcription factors, but disagrees for the direct regulation interactions. To substantiate author's claim, evidence of direct regulation and plausible physiological data is needed. There may be an idea of splitting this manuscript into several manuscripts, such as genetic interaction part and individual transcription factors, respectively.

Response: We appreciate the reviewer's concern. To the reviewer's former comment, we have employed multiple tests consistent with the most rigorous standards in the field. This includes: (1) Screening each Hfr donor transcription factor (TF) gene mutant twice against an arrayed F-recipient single gene deletion mutants in quadruplicate, generating eight replicate pairs per donor for statistical reproducibility. (2) Scoring genetic data by multiplicative and machine learning-based Gaussian process models, and considered only those genetic interactions (GIs) present in both scoring models as reliable. (3) Supporting high-confidence GIs by multiple lines

of computational and experimental evidence, including biological replicates and different assays, particularly wherever necessary in a concentration dependent manner.

Regarding the reviewer's next point, whether the regulation of a TF to a regulon of TF of another GI pair is direct or indirect warrants detailed characterization on a case-by-case basis. The reason being the underlying mechanism of GIs between TF gene pair is quite complex as TFs displaying strong GIs can regulate a number of genes themselves. Nevertheless, to address the reviewer's point, we hypothesized the likelihood that GI between a pair of TFs can be explained indirectly by the interactions between their targets. To examine this, we analyzed GIs of the predicted targets from RegulonDB database for each pair of TFs. As with the findings in yeast¹, TF pairs with strong GIs were not enriched for GIs between their targets (**Supplementary Fig. 2c**). One-third of the TF pairs (31.8%, 8,200 of 25,749) from rich media (RM) and minimal media (MM) static networks displaying significant GIs have at least one pair of targets with a stronger GI, which is lower than what is observed (36.8%; 9,500 of 25,749) from randomized data. Taken together, this data establishes that indirect effects of the targets cannot be the only reason for the large number of GIs observed between TFs. We have now included this data in the revised version to strengthen the relevance.

To the reviewer's latter comment on splitting this manuscript into several manuscripts, we reiterate that the initial goal of this study was to generate a useful resource for the broader community to motivate hypothesis-driven research. Based on this notion, we focused on addressing what genetic interdependencies among bacterial TFs are being altered in unstressed (i.e., static) or in the presence of environmental perturbation (i.e., differential), and how loss-of-function of a TF is sensed and regulated within the cell. Although our genetic maps allowed to gain better understanding of global epistatic relationships between components of the TF machinery that are altered by environmental distress, to showcase the utility of our static and differential genetic maps, we focused on three case studies on unannotated TF genes not previously linked to cellular processes. Upon characterizing them experimentally, we revealed their role in the regulation of cell division, cold shock adaptation, as well as functional dependencies mediating putrescine pathway and efflux pump. Besides characterizing these unexpected sub-systems, we believe our findings will open up exciting new avenues with many detailed follow-ups to explore on individual TFs in the years to come.

Comment # 3: To evaluate the data of genetic interactions between transcription factors, GI scores were calculated. Based on the GI scores, the authors picked up the several case and illustrated in Fig. 2, such as *crp-fis*, *hns-stpA*, and *baeR-cpxR-pppF*. Are these GI scores significant compared to the others? For example, in the case of *crp-fis* in RM, P-value of MP score is less than 4000th, and P-value of GP score is less than 5000th. Does this mean that more than 4000 pairs of transcription factors exhibit genetic interaction? Why did the authors select these pairs, why didn't the authors explain the more plausible GI score? Can the authors objectively assess the GI score and the genetic interaction between transcription factors? Throughout the manuscript, don't be arbitrary, the authors need to objectively explain why they picked up the GI with the GI scores compared with other GI scores including *yjdC-dicA*, *yneJ-acrR* and *ydiP-cspS*.

Response: We thank the reviewer for raising these discussion points. With respect to the reviewer's former question on selection of candidate gene pairs in **Fig. 2**, we note that the intent is to highlight whether our global epistatic relationships among TFs can be utilized to address the regulatory logic controlling individual TF genes. To highlight that the GI maps indeed have captured such regulatory relationships, we identified few examples to showcase regulatory modules involving gene pairs where one TF represses the function of another TF that acts as an activator, or two different TFs that function cooperatively or redundantly to regulate a distinct set of genes. In fact, the candidate gene pairs were selected based on literature evidence supporting their regulatory role to associate with their GI patterns from

static RM network. Most of these gene pairs also passed the set thresholds corresponding to two standard deviations in both scoring models, which we considered as a valid GI. We further point out that our quantitative assignment of GI score based on colony size measurement for each digenic mutant using either one of the scoring models were not ranked based on *P*-value. Instead, it is to help interpret the changes in double mutant fitness compared to random.

To the reviewer's next question on objectively assessing GI score vs. GIs between TFs, we have performed many quality metrics in an unbiased fashion in our original submission. For example, comparing static TF networks to previously reported GIs showed significant agreement by random sampling (**Supplementary Fig. 1e**). Likewise, genetically interacting TF genes present within the same functional domain or transcriptional regulatory family, and within the same operon in *E. coli* had significantly ($P \leq 0.05$) more positively correlated genetic profiles (**Supplementary Fig. 1f, g**) as pairs of co-expressed/co-transcribed TFs (**Supplementary Fig. 1h, i**) compared to random or other gene pairs.

To the latter part of reviewer's concern regarding the selection of *yjdC-dicA*, *yneJ-acrR* and *ydiP-csp* gene pairs, we specified in our original submission that static or differential networks were explored to find new functional roles for TFs based on their GIs with annotated genes. To address this, we followed-up on three representative set of TF gene pairs because: (1) *yjdC-dicA* ranked top 1% among all GI correlated profiles in static RM network, and exhibited a phenotype for further characterization. (2) *yneJ-acrR* ranked top 2 in static MM network among all *yneJ* correlated genetic profiles. While the goal is to characterize the physiological role of YneJ, we sought to comprehend how the aggravating GI of *yneJ* with *acrR* is linked to Gln-Glu and putrescine utilization (Puu) pathways. (3) *ydiP-csp* gene pairs ranked top 1% in static MM network among all *ydiP* correlated GI profiles. Notably, *ydiP* is also one of the hub gene identified in the differential network that not only showed strong GIs in MM compared to RM static network, but also exhibited a filamentous phenotype to pursue further. We have now incorporated some of these discussion points in the revised text for overall clarity.

Comment # 4: The transcriptional interaction between *dicA* and *yjdC* is unclear.

Response: The positively correlated GI profile between *yjdC* and *dicA* led us to investigate more into this observation by assessing cell morphology and transcript level changes between *yjdC-dicA* double and single mutants as outlined below and in detail in our original submission. To further ensure whether functional dependency between *yjdC* and *dicA* from genetic screen is legitimate, we confirmed independently this interaction by liquid culture growth assays (**Fig. 3f**), where unlike *yjdC* single mutant, *dicA* hypomorphic allele showed reduced growth in RM, and overexpression of *yjdC* rescued the growth defect of *dicA* mutant in a similar manner to *yjdC dicA* double mutant. The *yjdC-dicA* GI that was further elaborated (see response to this reviewer concern Comment #4.1) with a new ChIP-qPCR experiment show *yjdC* binding in the vicinity of *dicA*, suggesting a possible regulatory logic.

Comment # 4.1. The authors claim that *yjdC* is regulated by *DicA*, and regulates *dicA*, *dicC*, and *dicB*. However, only transcriptional levels of the test gene were measured, and indirect effects could not be ruled out. To understand the transcriptional network, the proof of direct regulation is mandatory.

Response: We thank the reviewer for this appraisal. As requested, ChIP-qPCR was performed to ascertain whether YjdC binds to *dic* locus. In comparison to untagged control strain, the chromosomal FLAG₃-tagged YjdC grown in RM and MM showed a 1.4- to 8.2-fold enrichment of YjdC occupancy at *dicABC*, respectively, implying *dic* alleles are likely targets of YneJ. The revised text based on the new results is now presented in **Fig. 3h**.

Comment # 4.2. The expression levels of *dicB* and *dicC* were increased in *dicA yjdC* double knockout strain. Thus, the formation of filamentation in *dicA* mutant was occurred by *dicB* and *dicC* expression, *dicA yjdC* double knockout strain should form the filamentation, but not.

These data indicate that the reason of filamentation is not only the expression level of dic genes, and it suggest that there are other yjdC target genes involved in formation of filamentation, or YjdC senses the environmental change which occurred by dicA regulon.

Response: We thank the reviewer for raising these discussion points. Though we cannot rule out the possibility of other *yjdC* targeting genes involved in the formation of filamentation, we have not tested such scenario based on the *yjdC* genetic profile we gathered (**Supplementary Table 4**), and it warrants additional investigation, which is beyond the scope of the manuscript. But we can draw conclusions based on the evidence we have at hand to pinpoint why *yjdC* might be involved in the regulation of filamentation. First, we chose to investigate *yjdC-dicA* as it was the top-ranked gene pair with a high positive correlation score in static RM network compared to all other genetic profiles (see response to this reviewer concern Comment #3, **Fig. 3e** and **Supplementary Table 4**). Second, unlike *yjdC*, the *dicA* or *dicC* mutant as expected^{2, 3} formed extensive filamentation (~9.2 to ~6.7 μm avg. cell length). Strikingly, the *dicA* or *dicC* phenotype was rescued to near wild-type (~2.4 μm) only when *yjdC* was additionally deleted. Overexpression of *yjdC* in trans in wild-type, and *dicA* or *dicC* mutant exhibited filamentation, albeit to reduced cell length (~2.1 to ~4.1 μm) than *dicA* or *dicC* single mutant (**Fig. 3g**). Third, while *dic* transcript levels were not significantly affected in *yjdC* mutant compared to wild-type (**Fig. 3i**), overexpression of *yjdC* in a wild-type or *dicA* mutant led to a 3-to-12-fold reduction in *dicAB* transcript levels than wild-type, with no significant impact on *dicC* expression. This suggests that *yjdC* downregulates *dic* locus when expressed *in trans*, while *dicA* largely controls *dic* locus under wild-type conditions. Collectively, these data support a model (**Fig. 3j**) in which *dicA* activates *yjdC*, *yjdC* represses *dicC* in a *dicA* null allele, and *yjdC* act as a repressor of *dicA* and *dicB* when *yjdC* is overexpressed. To ensure that the control of *dic* locus does not irreversibly switch to *dicABC* regulation, *yjdC* binds to *dic* locus, facilitating *dicABC* expression to resume cell division. We have now included these points in the revised text.

Comment # 4.3. This reviewer does not think that there is no functional interaction between dicA and yjdC.

Response: Based on our responses to this reviewer Comments # 4, 4.1 and 4.2, functional dependency does exist between *yjdC* and *dicA*, which is supported by their genetic profile, liquid culture growth assays, and other aforesaid experiments.

Comment # 5: The transcriptional interaction between yneJ and acrR is unclear.

Response: As we pointed out in our response to this reviewer Comment #3, we aim to study the physiological role of YneJ, and how its epistatic connection with *acrR* that is positively correlated and ranked top 2 in static MM network among all *yneJ* correlated genetic profiles is linked to Gln-Glu and Puu pathways. Thus, we discussed in our original submission that in addition to confirming *yneJ-acrR* aggravating GI by liquid culture growth assays, *yneJ* interacts physically with *acrR* by co-immunoprecipitation and bacterial two-hybrid experiments. This is consistent with previous reports indicating that GIs arising between genes that encode physically interacting proteins tend to share a functional relationship^{4, 5}. As an independent confirmation of GI, we challenged both *yneJ acrR* double and single mutants to SDS (sodium dodecyl sulfate; a substrate of AcrAB) treatment. Loss of *yneJ* increased sensitivity to SDS, which was restored by overexpressing *yneJ* *in trans*. Conversely, *acrR* mutant exhibited significant resistance ($P \leq 8.0 \times 10^{-3}$), implying that endurance to SDS is conferred through increased activity of AcrAB efflux pump. This finding is supported by reversing the resistant phenotype of *acrR* mutant by overexpressing *acrR* *in trans* (**Fig. 4i**). As with *acrR* mutant, *yneJ acrR* double mutants were also SDS-resistant, whereas overexpression of *acrR* in the corresponding double mutants displayed a similar SDS sensitivity to *yneJ* single mutant, suggesting that *yneJ* is likely to activate *acrAB* efflux pump by repressing AcrR regulation. This notion was verified by ChIP-qPCR, showing a 10- to 19-fold enrichment of YneJ

occupancy (**Fig. 4e**) upstream of *acrR* (i.e., intergenic region of *acrA-acrR*). Also, consistent with RNA-sequencing (**Supplementary Fig. 4b**), qRT-PCR showed *acrAB* transcript level to be reduced by 2-fold, while *acrR* was enhanced by 2.4-fold upon *yneJ* deletion compared to wild-type strain in M9-tryptone. However, overexpression of *yneJ in trans* restored *acrAB* levels in *yneJ* mutant, while *acrR* transcript level was near to wild-type (**Fig. 4j**). Taking all supporting evidence, these experiments do illustrate the genuine biological informativeness of the GI between *yneJ* and *acrR*, as well as *yneJ* regulation of *acrR*.

Comment # 5.1. *Why did the yneJ acrR double mutant show growth defects under rich medium and minimal medium in GI screening? There is no clear explanation based on the features of yneJ and acrR.*

Response: In our original submission, we emphasized that under stress and environmental cues or changes in nutrient availability, the transcriptional regulator *acrR* acts as a local repressor of *acrAB* efflux pump by fine tuning *acrAB* transcription and averting undue *acrAB* expression, which otherwise could lead to a loss of membrane potential that is deleterious to *E. coli*⁶. While *acrR* senses many toxic metabolites and changes in environmental conditions, inhibition of *acrAB* by *acrR* would accumulate the metabolites. Thus, inactivating *acrR* triggers *acrAB* initiation to restore homeostasis and increase bacterial fitness under stress⁶. Since the transcription of *acrAB* is also controlled by activators such as *marA*, *soxS* and *rob*⁶, it is conceivable that *acrR* and *acrAB* are regulated by additional TF such as *yneJ*. We confirmed this notion in the revised version by fluorescent polarization assay (**Supplementary Fig. 4g,h**), where YneJ binding to *sad* regulatory region was enhanced by AcrR than with YneJ without AcrR or with *leuO* negative control, suggesting that YneJ binds to *sad-yneJ* intergenic region efficiently in the presence of AcrR. This functional connectivity between *yneJ* and *acrR* can be related to the growth defect observed in RM and MM, where it is conceivable that deletion of either *yneJ* or *acrR* activates *acrAB* pump to efflux toxic metabolites from Gln-Glu or Puu pathways, which mirrored with no growth fitness defect. However, deletion of *acrR* and *yneJ* exacerbates slow growth (**Supplementary Fig. 4c**), supporting the idea that these TFs function redundantly to regulate *yneI* operon required for the maintenance of cell viability as *sad* mutant impair growth phenotype due to operon failure⁷.

Comment # 5.2. *The authors claim that YneJ regulates sad-glsB-yneG operon and yneJ mutant showed growth defect under Gln, Glu, and Putr as a sole carbon source (Fig.S3a). For Glu and Putr catabolism, there are isozymes shown in Fig. 4D. Are these isozymes also under regulation of YneJ?*

Response: This is a valid question, and we thank the reviewer for asking for clarification. As we depicted in our original submission, Gln, upon uptake into *E. coli* is converted to L-Glu by glutaminase gene, *glsB*. The Glu is then decarboxylated to γ -aminobutyric acid (GABA) by Glu decarboxylase, *gadAB*, and feeds into the Puu pathway (**Fig. 4c**), where putrescine imported through *puuP* is γ -glutamylated by *puuA*. The ensuing γ -glutamylputrescine to GABA by *puuBCD* is transferred by GABA aminotransferase (*gabT*, *puuE*) to succinic semialdehyde, which is oxidized by succinic semialdehyde dehydrogenase (*gabD*, *sad*) to yield succinate for the tricarboxylic acid cycle⁷. We have shown that *sad-glsB-yneG* mRNA levels (**Fig. 4g**) were lower in *yneJ* mutant, while *yneJ* overexpression in *yneJ* deletion mutant strain restored mRNA transcript near to wild-type, confirming *yneJ* is an activator of *yneI* operon.

Our ChIP-seq data revealed YneJ binding upstream of Glu decarboxylases (*gadAB*), *puu* gene cluster (*puuABCD*), GABA aminotransferases (*puuE*, *gabT*), as well as succinic semialdehydes (*sad*, *gabD*) in M9-putrescine and/or tryptone media just below the ChIP-seq peak-calling analysis threshold. However, affinity-purified, endogenously FLAG₃-tagged YneJ when grown under Luria-Bertani (LB) and M9-putrescine/tryptone medium, followed by mass spectrometry, co-purified with members (e.g., GlsB, GadB, PuuACE, GabT, Sad, GabD) of

Gln-Glu and Puu pathways (**Supplementary Fig. 4a**). The mRNA levels for genes in this pathway, except the *yneI* operon and *gabD*, measured by RNA-sequencing were higher in *yneJ* mutant than wild-type strain grown in M9-tryptone (**Supplementary Fig. 4b**). This suggests *yneJ* regulates Puu genes by activating *sad-glsB-yneG* or *gabD* expression, while repressing others. Since *yneJ* (**Supplementary Fig. 3a**) or *sad* mutant exhibit growth defect in putrescine but not *gabD* mutant⁷ implies that *sad* is vital than *gabD* when *E. coli* utilizes putrescine as a carbon source⁷, and that *yneJ* seems to explicitly regulate *yneI* operon in putrescine-inducible pathway. We have discussed these points in the revised version.

Comment # 5.3. The expression levels of yneJ, sad, glsB, and yneG were increased in M9+Gln/Glu/Putr/Try compared to rich medium (Fig. 4E). Why compared it to rich medium? yneJ, sad, glsB, and yneG genes can be increased in minimal medium. To show the importance of metabolites, the control medium must be M9 medium containing other carbon source (glucose). If say further, similar induction levels were observed in M9+Tryptone medium, suggesting that induction does not appear to be specific by Gln/Glu/Putr. yneI and yneH had been renamed to sad and glsB, respectively.

Response: We appreciate this concern, which we have addressed by measuring the transcript level of *yneJ* and genes within the *yneI* operon in a wild-type strain grown in M9 carbon limiting growth medium containing the metabolites (Gln/Glu/putrescine/tryptone) vs. M9 medium containing glucose as a carbon source by qRT-PCR. Notably, *sad-glsB-yneG* and *yneJ* expression level was elevated from 1.6- to 3.7-fold in M9-Gln/Glu/putrescine/tryptone than in M9-glucose medium (**Fig. 4d**), implying as with Sad⁷, YneJ utilizes Gln/Glu/putrescine/tryptone as a lone carbon source. To the reviewer's latter question on renaming *yneI* to *sad* and *yneH* to *glsB* is a valid point. In our original submission, we did mention that "The *yneJ* gene is located close to sigma E (σ^E or RpoE) promoter region of *yneI* operon, encoding succinic semialdehyde dehydrogenase (*yneI* or *sad*), glutaminase (*yneH* or *glsB*), and a DUF4186 domain-containing orphan, *yneG*, of unknown function (**Fig. 4c**)". However, we do agree that we did not carry out this gene nomenclature for *yneI* and *yneH* throughout the manuscript. We have rectified this omission in the revised version of the main text as well as in **Fig. 4, Supplementary Fig. 3a and 4a, b**.

Comment # 5.4. In the ChIP-seq screening, the YneJ binding sites were altered between M9-putrescine and M9-tryptone medium. Under these two conditions, the target sequences were completely different (Table S8). What is the mechanism by which the target sequence of a transcription factor is altered in different media? The most suspicious is that the promoter region of sad-glsB-yneG operon could not be detected. The authors state that they set the cut-off and removed non-specific peaks in ChIP-seq (P16 L9). Along this line, YneJ binding level on sad detected by qPCR is also non-specific.

Response: We thank the reviewer for their thoughtful comments, as we feel it has given us the opportunity to clarify these important points. In the **Supplementary Table 8** of our original submission, we show 11 significantly enriched YneJ-binding sites in M9-putrescine and 7 in M9-tryptone media that passed the stringent peak-calling analysis ($P \leq 1 \times 10^{-4}$). This does not mean the undetected YneJ-binding sites in one of the tested media growth conditions is due to their target sequences being altered but rather they were not shown because they fell below the chosen cut-off and were not statistically significant. In fact, all of the 18 reported YneJ-binding sites were detected in both conditions, but in one of the conditions they were below the select cut-off and hence they were not shown. With respect to removing non-specific peaks in ChIP-seq, what we meant is that prior to the cut-off selection, we retained a total of 1,891 YneJ-binding sites from either one of the experimental growth conditions after removing non-specific peaks from YneJ that overlapped with regions in control (i.e., 'input' and 'mock' ChIP with immunoglobulin G antibody) ChIP-seq experiments. Over one-third (40%, 748 of 1,891) of the YneJ-binding sites were detected in both conditions below the cut-off, including *sad*. Whereas,

the rest comprising, *glsB* or *yneG*, was detected only in M9-putrescine or tryptone medium, consistent with binding properties varying on growth conditions^{8,9}.

Just because we were unable to detect *sad-glsB-yneG* should not be interpreted as a failure of our methodology, but it reflects stringent criteria used to reveal strong binding sites. However, to ensure the *sad* regulatory region is a genuine YneJ binding site and not from noise, we performed ChIP-qPCR on an endogenous FLAG₃-tagged YneJ in M9-tryptone or putrescine media, revealing a 2.0- to 2.6-fold enrichment of YneJ occupancy upstream of *sad* (i.e., *sad-yneJ* intergenic region; **Fig. 4e**) compared to YneJ-FLAG control strain grown in M9-glucose. In fact, the ChIP-qPCR assay is proven to be an independent validation method for ChIP-seq, allowing for analysis of more independent replicates with proper control strains, and reducing artifacts introduced during ChIP-seq library amplification¹⁰. Thus, *yneJ* binding in *sad* gene confirmed by ChIP-qPCR is not non-specific.

Comment # 5.5. *YneJ recognition sequence in sad promoter is suspicious. In Fig. S3c, surrounding sequence is also conserved, and the highlighting sequence is not specific. The resolution of ChIP-seq or ChIP-qPCR is hard to determine the consensus sequence. Is there any consensus sequence among the YneJ targets in Table S8?*

Response: This is a valid question. The highlighted consensus sequence of *yneI* operon and *yneJ* to γ -Proteobacterial species in **Supplementary Fig. 3c** is based on the enriched (*E*-value = 0) sequence motif revealed by MEME suite (**Supplementary Fig. 3d**). In fact, the sequence shown in **Supplementary Fig. 3c, d** is identical to the one shown in **Fig. 4b**, except that the consensus sequence (TTCTCGATTCGTGAA) is reverse complemented. Besides, we tested whether the identified consensus motif is the genuine YneJ binding site that is crucial for *sad* regulation by fluorescence polarization. Our results suggest that YneJ in fact strongly binds to the conserved site in the *sad-yneJ* intergenic region at high affinity with Gln (**Fig. 4f**). That said, reviewer is right to inquire as to why some consensus sequence though conserved (**Supplementary Fig. 3c**) were not pursued. This was an oversight from our end, and we should have presented this more clearly. The motif-based sequence analysis in fact revealed three enriched (*E*-value = 0) conserved motifs (**Supplementary Fig. 3d**), but we examined only one of the three motifs by fluorescence polarization. We found the third consensus motif (CCAGTTGGGTCAGAT) to be present within the *yneJ* gene. However, given that the second motif (AATATTATTCATTTT) is within the promoter region of *yneI* operon, and in very close proximity to the motif we tested, we anticipate similar outcome from the second motif as in **Fig. 4f**. To address the latter part of the reviewer's concern, we did not observe any consensus motif when DNA sequences of YneJ targets shown in **Supplementary Table 8** were aligned. All these details were now included in the revised main text and supplementary figures.

Comment # 5.6. *In gel shift assay (Fig. 4g), non-specific binding has not been ruled out. Another DNA sequence should be tested for negative control. Effector test should be performed in different concentration manner. Other metabolites (at least Glu and Putr) should be tested to show the specific effect of Gln.*

Response: We thank the reviewer for raising this comment. In our original submission, we did rule out the non-specific binding by including a 28-bp oligonucleotides designed within the promoter region of *leuO* gene that is supposedly not to bind to *yneJ*, in the presence or absence of Gln, based on ChIP-seq data, which served as a negative control. Regarding the reviewer's next point, we did perform the fluorescence polarization assay with 21-bp fluorescent-labeled oligonucleotides as DNA probes with increasing concentration of YneJ recombinant protein at various Gln dilutions (i.e., 2 mM, 100 μ M, 2 μ M, 100 nM, 25 nM, 2nM). We found polarization to be increased with increasing concentration of YneJ protein with the addition of Gln (100 nM), and this trend appeared to be consistent when Gln was tested at all different amounts. To the reviewers last question, we have now performed the fluorescence polarization

experiment as in **Fig. 4f**, but with other metabolites (Glu, putrescine) at two different concentrations (2 mM, 2 μ M). While YneJ binding to *sad* was improved in the presence of Glu and putrescine at all concentrations tested than in the absence of these metabolites or with *leuO* negative control gene, the efficiency of YneJ binding to *sad-yneJ* intergenic region was robust with Gln than Glu or putrescine. We have incorporated these points in the revised version, and as there are too many curves, we took a representative metabolite concentration (i.e., 100 nM Gln and 2 mM Glu or putrescine) to portray in **Fig. 4f** and **Supplementary Fig. 3f**.

Comment # 5.7. *To test the yneJ regulation, the effect of Gln addition should be tested. Why sad-glsB-yneG did not induce in rich medium (Fig.4e)? Gln is included in rich medium.*

Response: In the **Fig. 4d** of our revised submission, we measured the transcript levels of *yneJ* and genes in *yneI* operon in MM containing Gln, Glu, putrescine, and tryptone and compared it to M9-glucose (see response to this reviewer concern Comment #5.3). Next, to this reviewer's question as to why genes in *yneI* operon did not induce in RM. This is because the expression level in RM was normalized to 1 to calculate the relative fold change between MM containing the metabolites vs. RM growth condition. Hence, it appears as though that the transcript levels were not induced in RM, which is not the case based on the aforesaid reasonings.

Comment # 5.8. *In Fig. 4f. The control should be set as YneJ-FLAG under M9+glucose. The untagged sample is just an experimental control. Also, there is no control for input (background) (Fig. S3f).*

Response: We apologize for this oversight, and, accordingly, have now show YneJ-FLAG under M9-glucose (relabelled to **Fig. 4e** in revised version) as opposed to untagged strain as an experimental control. Regarding the reviewer's next point, we have now included the input control for both immunoprecipitates and whole cell extracts. The amended **Supplementary Fig. 4d** (relabelled from the original submission) is now presented in the revised submission.

Comment # 5.9. *What happens to AcrR that forms complex with YneJ? If the DNA binding affinity changes, it can be tested in vitro, such as by using gel shift assay.*

Response: As requested, we have now performed the fluorescence polarization assay with increasing concentrations of YneJ protein in the presence (0.5 and 1.0 μ M) and absence of AcrR using the same 21-bp fluorescent-labeled oligonucleotides of the consensus motif in *sad-yneJ* intergenic region (**Fig. 4b**). Polarization increased considerably with YneJ protein when AcrR was added at both concentrations (0.5 μ M data shown; **Supplementary Fig. 4g, h**) than YneJ with out AcrR or with *leuO* negative control to rule out non-specific binding. This suggests that YneJ binds to *sad-yneJ* intergenic region efficiently in the presence of AcrR. We have now included this new finding in the revised version of the main text.

Comment # 5.10. *acrR mutant did not show the growth difference against SDS stress in Fig. 4i. This result suggests that acrR did not involve in SDS resistance? AcrR is repressor of acrAB operon. If YneJ acts as a repressor for AcrR, SDS sensitivity reduces in yneJ mutant, but the result showed opposite effect.*

Response: We agree on the former point, and have now repeated the growth assay with more replicates to explain the growth difference of *acrR* mutant in the presence and absence of SDS. In fact, *acrR* mutant exhibited significant resistance ($P \leq 8.0 \times 10^{-3}$) when compared to growing the respective mutant strain without SDS, implying that endurance to SDS is conferred through increased activity of AcrAB efflux pump. This finding is supported by reversing the resistant phenotype of *acrR* mutant by overexpressing *acrR* in trans (**Fig. 4i**). We have included these points in the revised main text to strengthen the relevance. To the reviewer's latter question, the confusion might have occurred from the color schema used in **Fig. 4i** because in our original submission we did show that the loss of *yneJ* is sensitive to SDS as the reviewer pointed out.

Comment # 5.11. *Fig. S3b. Is AcrR also conserved between YneJ conserved strains?*

Response: Phylogenetic analysis on the orthologs of *acrR* across 54 closely related γ -proteobacteria indicated that only one-ninth (11%; 6 of 54) of the species were conserved, suggesting that in contrast to genes in *yneI* operon and *yneJ* (**Supplementary Fig. 3b**), *acrR* had a low-degree of sequence co-conservation ($p = 9.3 \times 10^{-2}$; **Supplementary Fig. 4f**). We have now added the findings from this analysis in the revised main text.

Comment # 5.12. The author's idea that yneJ responses to compounds in putrescine degradation pathway was derived from the function of sad-glsB-yneG operon. The gene names of sad and glsB must be used.

Response: We have made the correction.

Comment # 6: The transcriptional interaction between ydiP and CSPs is unclear.

Response: In our original submission, we pointed out that the differential interaction hubs (i.e., genes with many interactions) modulate a variety of cellular functions¹¹, and to explore one of the hubs in differential network, we chose *ydiP*, an uncharacterized HTH-type transcriptional regulator that showed strong GIs in MM (35%; 41 of 116) than RM (16%; 19 of 116; **Fig. 6a**), and were not previously linked to the CspA family of cold shock protein (CSP) encoding genes (**Fig. 6b**). We show that *ydiP* genetic profiles is positively correlated with *csp* genes (**Fig. 6c**). Using affinity-purification of YdiP and mass-spectrometry, we found YdiP to be physically associated efficiently with 5 (CspABEGI) of the 9 CSPs in RM, and only with CspE in MM at 15°C (**Fig. 6d**), but not under both cultures at 37°C. In addition, the physiological role of YdiP at low temperature was examined by several assays, which led to the conclusion that *ydiP* as a regulator (**Fig. 6j**) repress and activate various CSPs under RM and MM culture conditions, respectively, at low temperature. Therefore, considering all the evidence presented in **Fig. 6** and **Supplementary Fig. S6**, there is a general consensus that *ydiP* and CSPs are functionally dependent, which we have further elaborated in our responses to this reviewer's concerns below.

Comment # 6.1. Why did ydiP GIs show opposite effect between cspACG and cspFDH? This effects were inconsistent with growth test in Fig S5b (P23L18). Why? Furthermore, in Fig. 6g and Fig. S5d, the cell form of ydiP was restored by causing double mutants with either csps gene. This effect does not match the GIs effect.

Response: This is a valid point. With respect to the former, YdiP interact with members of the CspA family at different levels by displaying differential aggravating and alleviating GIs with *cspACG* and *cspFDH*, respectively (**Fig. 6b**). This suggests that YdiP may have specialized functional roles with CspA family members that typically regulate differently¹². Nevertheless, the strong positive correlations (**Fig. 6c**) and GIs observed between YdiP and all CSPs (**Fig. 6b**) imply that the functions of these proteins are closely related in a broader cellular context, and share functional connections. Regarding the reviewer's next point, the discrepancy between **Fig. 6b** and **Supplementary Fig. 6b** in the original submission is that the GI screens were conducted in RM and MM at 32°C. The reason for choosing 32°C is that temperature-inducible λ Red cassette in strain DY330¹³ marked with Amp^R (resistance to ampicillin) was moved by P1 transduction into Hfr Cavalli. Since λ Red cassette is under the control of a temperature-sensitive repressor, we incubate all the donor gene deletion mutant strains at 32°C. Also, to minimize growth variability between donor and recipient mutants for genetic screening, we incubate all the plates at 32°C. However, growth assay was performed in RM at 37°C and 15°C. As we mentioned in our original submission, the rationale for doing this way is not to confirm GIs but to understand YdiP role and its association with CSPs at low temperature. To address this, growth rates were measured upon a temperature shift from 37°C (used as a control for comparison) to 15°C. As with previous report¹⁴, *ydiP* and *csp* mutants showed no difference in growth rate at 37°C when compared to wild-type. But the defect was exacerbated when *ydiP* and *cspABEGI* were deleted, but not the *ydiP cspCDFH* double mutants which resembled one of the single mutants (**Supplementary Fig. 6b**), indicative of functional dependency.

To the last question, in contrast to 37°C, *ydiP* mutant grown in RM or MM after 4 hrs at 15°C formed long filamentous cells (~6.1 µm avg. cell length) relative to wild-type (~2.1 µm), while *ydiP* overexpression in *ydiP* mutant rescued the filamentous phenotype. In the case of strains lacking both *ydiP* and *csp*, the filamentous cells were similar to one of the single mutants in RM or MM at 15°C, but not at 37°C (**Fig. 6g** and **Supplementary Fig. 6d, e**; data not shown for MM at 37°C). While this result suggests YdiP to target CSP-family proteins at low temperature, we cannot make direct comparison of the cell morphology to GI phenotype based on the models employed. This is because multiplicative or Gaussian Process models are useful for quantitative measures such as colony growth fitness or growth rate but less so for complex phenotypes like cell morphology, implicating that alternative models will be needed to assess epistatic behaviour^{15,16}. We have included some of these points in the revised main text.

Comment # 6.2. P24L1. *The authors claim that YdiP expression level was enhanced from 2hrs after shifting to cold shock, but in Fig 6e, YdiP expression levels were similar at 0hr and 2hrs.*

Response: We apologize for this oversight as this was an inadvertent error on our end when incorporating the quantification values. We have re-quantified the blot, and the corrected value after normalizing to *E. coli* Hsp60 loading control is now shown in **Fig. 6e**.

Comment # 6.3. P24L3. *The authors claim that ydiP transcript level was enhanced from 4hrs after shifting to cold shock, but YdiP expression levels were similar at 0hr and 4hrs.*

Response: We have made the correction. See our response to this reviewer's comment #6.2.

Comment # 6.4. P24L18. *Why the authors set the cut-off for 50 bp upstream and downstream from start codon in YdiP ChIP-seq screening? In the case of YneJ, they set 200 bp from start codon. In Table S12, What is the mechanism by which the target sequence of a transcription factor is altered in different temperature?*

Response: We apologize for the confusing nature of the sentence structure. For both YneJ and YdiP ChIP-seq experiments, peak-calling threshold was set to $P \leq 1 \times 10^{-4}$, which corresponds to genes with start codons from 200 bp upstream and downstream of a binding site that was considered as potential YneJ targets. Likewise, the $P \leq 1 \times 10^{-4}$ corresponds to 50 bp upstream and downstream of a binding site as putative YdiP targets. We have clarified this point in the revised version. Regarding reviewer's latter comment (see also our response to this reviewer's Comment #5.4), YdiP ChIP-seq was performed not at different temperature but at 15°C in two (RM, MM) different growth conditions. Our results suggest that one-seventh (15%, 9 of 61) of the YdiP-binding sites were enriched in both growth conditions and passed the set peak-calling threshold ($P \leq 1 \times 10^{-4}$). The rest that was undetected in one of the growth conditions is not due to their target sequences being altered but rather fell below the chosen cut-off.

Comment # 6.5. *The interaction between YdiP and cspE was shown, however, what about other csp genes? At least, it is not enough to prove that YdiP acts as a dual regulator for CSPs. Including cspE, functional interactions between ydiP and csps can be indirect effects.*

Response: We thank the reviewer for this assessment. With respect to the former, as we noted in the original submission, YdiP binding site was detected within the *cspE* TF-encoding gene in RM (i.e., 35 bp from the transcription start site) but not under MM at 15°C (**Fig. 6h**). Although YdiP binding site that was detected with other *csp* genes in RM (i.e., *cspABDEHI*) or MM (*cspBDI*) at 15°C were significant ($P \leq 0.01$), the binding sites were far (i.e., -545 to +95 bp) from the predicted transcription start site of *csp* gene, except *cspA* (+79 bp); however, they all missed the stringent threshold ($P \leq 1 \times 10^{-4}$) set in the peak calling analysis. Hence, we examined YdiP occupancy at *cspE* loci by ChIP-qPCR, which enhanced on an average by 12.2-fold in RM or 2.7-fold in MM at 15°C than in either growth conditions at 37°C (**Fig. 6h**). Notably, though below threshold by ChIP-seq, ChIP-qPCR showed a 7.1-fold increase of YdiP

occupancy at *cspA* loci in RM at 15°C than at 37°C (**Supplementary Fig. 6f**). These results suggest that CspAE loci are likely targets of YdiP at low temperature.

To the reviewer's next point, based on the evidence at hand from ChIP-seq, ChIP-qPCR and qRT-PCR experiments, we can conclude that *ydiP* regulates either by repressing or activating various *CSPs*, depending on RM or MM growth conditions at low temperature. But we do concur with the reviewer that the use of "dual regulator" may have been an overstep from our end, and for clarity we have reworded this sentence in the revised version. To latter part of the reviewer's concern, additional follow-up investigations will be warranted to understand whether the functional interactions observed between *ydiP* and *csp*s are due to direct or indirect effects. We have now added these points in the revised main text.

Comment # 7: *Identification of genetic interaction was performed under rich medium and minimal medium and many interactions showed different growth between media. However, little explanation is given based on the function of transcription factor. Thus, the credibility of the data is reduced.*

Response: In our original submission, we have provided evidence on how differential epistasis map can reveal new gene functions that were undetected in static GI networks. This includes:

1. Examination of the differential and static networks, pinpointing TFs with known roles in metabolism, transport, stress response, and those with unknown function that were more enriched in differential (**Fig. 5c**) than in the static RM and MM networks.
2. In **Fig. 5e** and **Supplementary Table 10**, we show that one-fifth of TF genes (20%; 55 of 282) had high autocorrelation (> 0.25) in RM and MM, while the profiles for one-tenth (10%; 28 of 282) of TFs had low autocorrelation (< 0.25). This observation has been linked with two examples, focusing on a global TF, *lrp*, a leucine-responsive regulatory gene involved in metabolic functions¹⁷, and *ydhB*, a novel LysR family regulator, whose role in TF that is less well understood, which showed low autocorrelation with GI patterns between conditions.
3. Many differential GIs between modules (i.e., complexes and pathways) that have not been previously linked to TF mechanisms, allowing to infer testable hypotheses. The latter was discussed with examples (**Supplementary Fig. 5f**). First, TF gene encoding for Crp and CytR (cytidine regulator) dimeric proteins, forming a nucleoprotein complex with opposing effects on transcription¹⁸ showed distinct differential GI patterns. Second, the module pair with distinct genetic profiles included the differential aggravating GI between *crp* and *fnr* global regulators.
4. Highlighting static genetic changes between RM and MM growth conditions by categorizing genes pairs with gain or loss of interactions (**Fig. 5f**). GIs underlying the changes in growth conditions were implicated with DNA-binding transcriptional regulator, *phoP* of the two-component system, that controls the transcription of an another two-component regulatory system, *rstA*, with a gain of interaction in MM compared to RM. Conversely, the global regulator, *crp*, that regulates the transcription of its target master motility complex, *flhDC* was shown with the loss of GI solely in MM relative to RM growth condition.
5. Investigating differential GI network to unveil one-fifth (22%; 66 of 302; **Fig. 5g**) of the TFs to be hubs (i.e., ≥ 100 GIs), and of those 47 were annotated to metabolism, transport and stress response, as well as to unknown function. As a case study, we examined one of the hubs identified in our differential network, *ydiP*, an uncharacterized HTH-type transcriptional regulator that showed strong GIs in MM compared to RM (**Fig. 6a**), and were not previously linked to the CspA family of cold shock protein encoding genes (**Fig. 6b**).
6. Enriched ($|Z\text{-score}| \geq 2$; $P \leq 0.05$) co-conserved interprocess gene pairs interconnected by GIs showed notable differences in static and differential networks (**Supplementary Table 15**), such as the enrichment for conserved differential GIs between stress response and transport or translation, as well as between metabolism and bacterial motility (**Supplementary Fig. 7a**).
7. Correlated GI patterns among TF paralog pairs in the static and differential networks, which revealed one-third of paralog pairs (34%; 68 of 201) with distinct set of GI profiles in response

to changing conditions (**Supplementary Fig. 7d**). For example, GI profiles among the paralogs of OmpR family of 14 response regulators (*arcA*, *baeR*, *basR*, *cusR*, *cpxR*, *creB*, *hprR*, *kdpE*, *ompR*, *phoB*, *phoP*, *qseB*, *rstA*, *torR*) were altered in MM and differential networks.

Besides these findings, we could expand on the function of TFs in detail but since the manuscript is already longer than usual, as rightly pointed out by Reviewer #3 in Comment # 2, we believe the aforesaid cases discussed in greater depth should exemplify the biological informativeness on the function of some of the TFs altered in different growth conditions.

Comment # 8: *P11L10, There is a publication showing that *ygfI* gene is involved in oxidative stress resistance*

Response: We thank the reviewer for bringing this to our attention, but we were unable to find the publication referencing *ygfI* gene to oxidative stress resistance. Nonetheless, we feel this is outside the context as we are not linking *ygfI* to stress response at a gene level, but rather associating how stress transcriptional regulators (*cspABE*, *oxyR*) are shared between *ygfI* and *yddM* by similar alleviating GIs, especially given the GI profiles of *ygfI* and *yddM* were highly correlated. Based on the association observed, we suggested a possible coordinated function of *ygfI* and *yddM* with cold shock (*cspABE*) and oxidative (*oxyR*) genes related to stress response.

Comment # 9: *P3L14, References 10 and 12 are not in vivo ChIP screening, in vitro genomic SELEX screening.*

Response: We have corrected it as suggested.

Comment #10: *To rename the gene name should be done carefully. The transcriptional role of *ygfI* and *yneJ* is still unknown.*

Response: Besides the data provided in the original submission, and with the inclusion of new data and other substantial changes to the revised version requested from this reviewer (see Comments #5 until #5.12) further strengthens the *yneJ* model. Briefly, our data supports that under carbon limiting condition, *yneJ* regulates the *yneI* operon, and activates *acrAB* pump by repressing *acrR* to efflux toxic metabolites from Gln-Glu or Puu pathways. Thus, we believe based on the biological significance it is appropriate to assign *yneJ* as *pggR* for putrescine and Gln-Glu gene regulation. Regarding *ygfI*, due to lack of supporting experimental evidence we did not make any claims in the original submission to rename this gene.

Reviewer #2:

Comment # 1: *This paper described the genetic interaction of 302 predicted transcription factors, including experimentally verified and unknown functions, in E. coli K-12 by systematic construction of double deletion strains by conjugation and analyzed on rich medium and minimal medium, or differential analysis when switching from rich medium to minimal. This group was an original member of the group that established the method for analysis of genetic interaction in E. coli in 2008 (Butland et al.), and have been continuously performing genetic interaction analysis in E. coli since then, by conjugation method to construct double knockout strains. The authors performed statistical analysis appropriately together with the structural and functional information of the transcription factors. In addition, some of the hypotheses obtained from the genetic interaction analysis were experimentally verified for a group of transcription factors with entirely unknown functions, showing that functional prediction is possible. The analysis performed is a large-scale analysis, and their analysis was carefully designed and performed using reliable methods. This paper showed a new way to elucidate the dynamic rewiring of the transcriptional network and should be beneficial for the readers.*

Response: We thank the reviewer for recognizing the value of this resource.

Comment # 2: In the last paragraph of the abstract, the authors mention pathogenesis, but there does not seem to be any particular analysis to pathogenesis in the analysis. Moreover, it seems too abrupt. Is it really necessary?

Response: We agree with the reviewer regarding the mentioning of “pathogenesis” in the abstract, and have now revised the text to reflect our observation more accurately.

Comment # 3: In the results, the authors described the 304 transcription factors targets, and out of these, 278 were converted to Hfr by the lambda RED recombination method. On the other hand, Supplemental Table 2 showed a list of interaction pairs, and Gene 1 and Gene 2 listed 301 and 300 genes, respectively. The recipient strains were clearly from the Keio collection, but the donor side strains were not clear. Therefore, preparing the table listed Hfr strains is clearer, including conditional deletion strains used in the experiment.

Response: We appreciate these concerns. We have generated digenic mutants in two different growth conditions (RM, MM) by conjugating 278 individual hyper-recombinant Hfr-Cavalli query ‘donor’ gene mutant strains marked with a chloramphenicol (Cm)-resistance cassette, against an arrayed 302 F- ‘recipient’ mutant strains marked with a kanamycin (Kan)-resistance cassette from Keio knockout strain collection. Each Hfr donor query TF gene mutant was screened twice against an arrayed F- recipient single gene deletion mutants in quadruplicate, generating eight replicate pairs per donor for reproducibility. Since each gene pair was tested in both ways, in total we had 16 colony size measurements. Additionally, data quality and consistency were maintained by applying the following filtering measures: (i) normalized colony size difference for a given gene pair between two replicates should not be > 0.5 , and (ii) the correlation coefficient for each gene pair between replicate screens should not be < 0.5 . Gene pairs that did not meet these criteria were removed from analyses. We also removed gene pairs with a chromosomal distance of 30 kbp on either side of the donor query loci for linkage effects that cause false positives¹⁹. Using multiplicative¹⁹ or Gaussian Process model²⁰, GI score was then computed for each replicate gene pair, and examined for similar GI phenotype (i.e., aggravating or alleviating). If the replicate gene pairs followed a similar trend in terms of GI phenotype, their GI scores were averaged to create one composite score. If there was an inconsistency with GI phenotype score measurements among replicate gene pairs (i.e., variance ≥ 0.05), then the gene pair with GI score closer to zero (or neutrality) was chosen as a conservative measure. Due to these reasons, it is not possible to display Hfr and F- strain information for each gene pair in the **Supplementary Table 2**.

To the reviewer’s other question, in **Supplementary Table 2** of our original submission, we have shown gene pairs scored either in both (RM, MM) or in one of the growth conditions. However, given that only viable double mutants from RM were replica pinned onto MM, GI scores were not generated in MM for synthetic lethal gene pairs tested in RM condition. In addition, the **Supplementary Table 2** includes certain gene pairs in RM that failed to pass the set statistical threshold corresponding to two standard deviations ($|Z\text{-score}| \geq 2.0$; $P \leq 0.05$) but were captured in the MM growth condition. Thus, we cannot simply tally the Gene1 and Gene 2 columns in **Supplementary Table 2** as it is not appropriate in this case because it simply implicates the GI scores for all possible number of gene pairs, encompassing 302 TFs, that were tested and scored in different growth conditions. We have now included some of these details for readers to understand how the analyses were performed.

Comment # 4: Page 5, line 10, is there any reference to the statement, “mazE-ala-tRNA has a significant effect on growth”? If yes, it should be cited.

Response: We thank the reviewer for this suggestion, and have now referenced prior work²¹ for non-essential deletion mutant (alanyl-tRNA synthetase *alaS*) but not the hypomorphic essential antitoxin *mazE* that had an effect on bacterial growth in our study.

Comment # 5: Page 5, line 14, In the case of transferring from RM to MM, the effect of changing the culture medium may differ significantly depending on the deleted gene. In particular, those related to trace elements are likely to be greatly affected by the change of the medium. What measures have the authors taken in this regard? Couldn't it have been done with MM from the beginning, including the conjugation step?

Response: We appreciate this comment. During the eSGA method development^{22, 23}, we found the use of RM for the conjugation step resulted in more consistent growth of F- recipient gene deletion mutant strains in a reasonable screening time frame. Cells grown in MM not only required much longer incubation times (about 2 weeks to complete an eSGA screen when pinned onto MM vs. <6 days using RM), but also displayed greater growth rate variability among different recipient mutants and poorer reproducibility between replicate screens. Furthermore, this variability leads to inconsistent and less efficient conjugation when a constant number of donor cells were mated with varying numbers of recipient cells (depending on the viability of each F- recipient mutant in MM). Hence, we opted for a more reliable screening plan to minimize spurious variance while simplifying the study logistics.

Comment # 6: Page 8, line19, crp deletion strains could not be purified in the Keio collection (Yamamoto et al.). How did the authors use crp deletion strains for this analysis? It seems highly likely that the strain is a partial diploid. Did the authors purify them?

Response: This is a valid question, and we thank the reviewer for asking for clarification. We are aware of the *crp* deletion mutant strain being partial diploid in the Keio F- recipient single gene knockout library. As a result, in this particular instance, we have created the *crp* mutant strain marked with kanamycin (Kan) in house in BW25113 parental strain. We have now incorporated this point in the updated methods section of the revised manuscript.

Comment # 7: Page 18, line 16, "other yneJ interacts physically with acrR, with which it showed an aggravating GI." Genetic interactions between genes, whose products form protein complexes, are thought to be alleviating, but this combination showed an aggravating interaction. However, this combination shows an aggravating interaction experimentally, so what is the reason for suspecting a physical interaction to be confirmed? Does it mean that YneJ forms a complex with AcrR but ultimately regulates the function of AcrR as a TF, forms a complex is not essential for the AcrR TF function? Is this what the authors considered?

Response: This is a valid question. Initial epistatic studies in yeast have shown that alleviating GIs can arise within protein complexes as second deletion in an already cooperated complex does not cause an additional fitness defect^{16, 24, 25}. However, genome-scale GI maps constructed over 23 million double mutants in yeast by Boone and colleagues have shown that the majority of aggravating and alleviating GIs occurs between, rather than within complexes and pathways, linking those that likely to work together or buffer one another, respectively^{26, 27}. The same group further suggested that alleviating GIs between nonessential genes in yeast overlapped with protein-protein interactions, albeit to a lesser extent²⁷. Thus, in the case of AcrR and YneJ, it is unclear whether it forms a nonessential protein complex. However, the existence of *yneJ-acrR* gene pair whose products physically interact and share an aggravating GI suggests that they possess a functional relationship, and that simultaneous perturbation of two genes physically interacting show a fitness defect that do not resemble their single mutants. We have now added these points to the revised version of the main text.

Comment # 8: Page 31, line 21, very high concentrations of Kan (50µg/mL) and Cm (34µg/mL) are used for the single copy drug resistance gene in the genome. However, in the next page 32, line 7, reasonably low concentrations of Kan(25µg/mL) and Cm(17µg/mL) for the low copy plasmid are used for the pACYC. High concentration of antibiotics might affect the frequency of partial diploid appearance, have the authors checked this concerning issue?

Response: With respect to the former, the suggested Cm (34 µg/mL) and Kan (50 µg/mL) concentrations have been consistently used by us in our previously reported GI studies by eSGA^{22, 23, 28, 29}. Similar antibiotic concentrations have also been employed by others during the creation of double mutants³⁰. During eSGA developmental process, we noted that merodiploidy (i.e., partially diploid cells) is a chance event that appears as a result of failure in some step of the recombination process³¹. In fact, the numbers of merodiploid should not increase whether we have lower or higher concentration of Kan-Cm. Given that large number of double mutant strains are handled during the eSGA screening, some mutants tend to overcome resistance at lower concentration of Kan-Cm. Thus, we applied Kan and Cm antibiotics at the specified concentrations in our eSGA genetic screens, and the resulting GI networks from this study (**Supplementary Fig. 1e**) or in our previously reported studies^{22, 23, 28, 29} when compared against the reference set of GIs from the literature were in agreement.

To the last question, overexpression of *acrR* in *yneJ acrR* double mutants marked with Kan-Cm was produced using a pACYC184 low copy plasmid with tetracycline-resistance marker. The reason for lowering Kan and Cm to half the dosage in this case compared to eSGA genetic screens is because we were able to detect resistant colonies to Kan-Cm even at levels below minimum inhibitory concentration. Besides, when handling such strains, we typically streak single colonies, and manually verify by PCR for the marker presence and gene deletions.

Comment # 9: *Figure 1, a) jΔ::Cm in the right figure seems to be a mistake for jΔ::Km.*

Response: We have corrected it as suggested.

Comment # 10: *Supplementary Fig. 1, c) left Aggravating graph, why is the second bar graph shown with green dots? Is it a mistake for red?*

Response: Yes, and we have corrected it as suggested.

Comment # 11: *Supplementary Fig. 4, b) Is this a different form of the analysis by Tong et al. as a graph? Please check if this is an appropriate citation.*

Response: Yes, the analysis was done differently using Tong et al., dataset. In **Supplementary Fig. 4b** of the original submission (now referred to as **Supplementary Fig. 5b** in the revised version), we show autocorrelated GI profiles of each TF gene between RM and MM static networks against single TF gene deletion mutant growth fitness sensitivity in MM containing different carbon sources from the phenotypic genetic screen (Tong et al³²).

Comment # 12: *Finally, regarding the gene ID of E. coli, the b number is one ID, but the ECK ID should be included since it has been adopted by the international E. coli community. Cooperatively developed annotation snapshot--2005. Nucleic Acids Res 34, 1-9 (2006). PubMed: 16397293.*

Response: We do concur with the reviewer, and have included ECK IDs as requested in the **Supplementary Table 1** of the revised version of the manuscript.

Reviewer #3:

Comment # 1: *The work is interesting as provides new insights in how the regulatory machinery globally adapts to two different and opposite conditions. The set of epistatic interactions identified in this work conforms a map that provides interesting data to improve our understanding of the organization of global gene circuits. It also provide new putative functions for uncharacterized regulators and analyzes the co-conservation of genes involved in epistatic interactions.*

Response: We thank the reviewer for their appreciation and acknowledging our effort.

Comment # 2: *The manuscript is well written, clear, despite a bit longer than usual but maybe justified given the amount of work involved. Data generated in the work is reported as supplementary material and in a web portal developed by the authors (<http://ecoli.med.utoronto.ca/eMap/TF>).*

Response: We thank the reviewer for the positive comment on the manuscript.

Comment # 3: *The authors investigate to what extent their epistatic interactions among TF are “generalized” in other bacteria. To answer this they use phylogenetic profiles constructed using COGs (clusters of orthologous groups). However, the co-conservation, via orthology, of the source and target genes comprising an epistatic interaction is not sufficient to guarantee the existence of an epistatic interaction. Authors must further clarify this point to the reader.*

Response: We agree on this, and to raise reader’s awareness we have discussed this point briefly in the methods section of the revised manuscript due to space constraint.

Comment # 4: *Supplementary figure 2, panel d: There is a typo in “Bacterial response”, it must be “Bacterial response”.*

Response: We have corrected the typo as suggested.

References:

1. Zheng, J. et al. (2010) Epistatic relationships reveal the functional organization of yeast transcription factors. *Mol. Syst. Biol.* **6**, 420 (2010).
2. Bejar, S. & Bouche, J.P. A new dispensable genetic locus of the terminus region involved in control of cell division in Escherichia coli. *Mol. Gen. Genet.* **201**, 146-150 (1985).
3. Bejar, S., Bouche, F. & Bouche, J.P. Cell division inhibition gene dicB is regulated by a locus similar to lambdoid bacteriophage immunity loci. *Mol. Gen. Genet.* **212**, 11-19 (1988).
4. Bellay, J. et al. Putting genetic interactions in context through a global modular decomposition. *Genome Res.* **21**, 1375-1387 (2011).
5. Bandyopadhyay, S., Kelley, R., Krogan, N.J. & Ideker, T. Functional maps of protein complexes from quantitative genetic interaction data. *PLoS Comput. Biol.* **4**, e1000065 (2008).
6. Ma, D., Alberti, M., Lynch, C., Nikaido, H. & Hearst, J.E. The local repressor AcrR plays a modulating role in the regulation of acrAB genes of Escherichia coli by global stress signals. *Mol. Microbiol.* **19**, 101-112 (1996).
7. Kurihara, S., Kato, K., Asada, K., Kumagai, H. & Suzuki, H. A putrescine-inducible pathway comprising PuuE-YneI in which gamma-aminobutyrate is degraded into succinate in Escherichia coli K-12. *J. Bacteriol.* **192**, 4582-4591 (2010).
8. Ireland, W.T. et al. Deciphering the regulatory genome of Escherichia coli, one hundred promoters at a time. *eLife* **9** (2020).
9. Myers, K.S. et al. Genome-scale analysis of escherichia coli FNR reveals complex features of transcription factor binding. *PLoS Genet.* **9**, e1003565 (2013).
10. Fitzgerald, D.M., Bonocora, R.P. & Wade, J.T. Comprehensive mapping of the Escherichia coli flagellar regulatory network. *PLoS Genet.* **10**, e1004649 (2014).
11. Bandyopadhyay, S. et al. Rewiring of genetic networks in response to DNA damage. *Science* **330**, 1385-1389 (2010).
12. Graumann, P., Wendrich, T.M., Weber, M.H., Schroder, K. & Marahiel, M.A. A family of cold shock proteins in Bacillus subtilis is essential for cellular growth and for efficient protein synthesis at optimal and low temperatures. *Mol. Microbiol.* **25**, 741-756 (1997).
13. Yu, D. et al. An efficient recombination system for chromosome engineering in Escherichia coli. *Proc. Natl. Acad. Sci. USA* **97**, 5978-5983 (2000).
14. Stokes, J.M. et al. Cold stress makes Escherichia coli susceptible to glycopeptide antibiotics by altering outer membrane integrity. *Cell Chem. Biol.* **23**, 267-277 (2016).
15. Mani, R., St Onge, R.P., Hartman, J.L.t., Giaever, G. & Roth, F.P. Defining genetic interaction. *Proc. Natl. Acad. Sci. USA* **105**, 3461-3466 (2008).

16. Beltrao, P., Cagney, G. & Krogan, N.J. Quantitative genetic interactions reveal biological modularity. *Cell* **141**, 739-745 (2010).
17. Peeters, E. & Charlier, D. The Lrp family of transcription regulators in archaea. *Archaea* **2010**, 750457 (2010).
18. Rasmussen, P.B., Holst, B. & Valentin-Hansen, P. Dual-function regulators: the cAMP receptor protein and the CytR regulator can act either to repress or to activate transcription depending on the context. *Proc Natl Acad Sci U S A* **93**, 10151-10155 (1996).
19. Butland, G. et al. eSGA: E. coli synthetic genetic array analysis. *Nat. Methods* **5**, 789-795 (2008).
20. Kumar, A. et al. A Gaussian process-based definition reveals new and bona fide genetic interactions compared to a multiplicative model in the Gram-negative Escherichia coli. *Bioinformatics* **36**, 880-889 (2020).
21. Kelly, P. et al. Alanyl-tRNA Synthetase Quality Control Prevents Global Dysregulation of the Escherichia coli Proteome. *mBio* **10**, e02921-19 (2019).
22. Babu, M. et al. Genetic interaction maps in Escherichia coli reveal functional crosstalk among cell envelope biogenesis pathways. *PLoS Genet.* **7**, e1002377 (2011).
23. Gagarinova, A. et al. Systematic Genetic Screens Reveal the Dynamic Global Functional Organization of the Bacterial Translation Machinery. *Cell Rep.* **17**, 904-916 (2016).
24. Collins, S.R., Schuldiner, M., Krogan, N.J. & Weissman, J.S. A strategy for extracting and analyzing large-scale quantitative epistatic interaction data. *Genome Biol.* **7**, R63 (2006).
25. Schuldiner, M. et al. Exploration of the function and organization of the yeast early secretory pathway through an epistatic miniarray profile. *Cell* **123**, 507-519 (2005).
26. Costanzo, M. et al. The genetic landscape of a cell. *Science* **327**, 425-431 (2010).
27. Costanzo, M. et al. A global genetic interaction network maps a wiring diagram of cellular function. *Science* **353**, aaf1420 (2016).
28. Babu, M. et al. Quantitative genome-wide genetic interaction screens reveal global epistatic relationships of protein complexes in Escherichia coli. *PLoS Genet.* **10**, e1004120 (2014).
29. Kumar, A. et al. Conditional epistatic interaction maps reveal global functional rewiring of genome integrity pathways in Escherichia coli. *Cell Rep.* **14**, 648-661 (2016).
30. Zhao, Z. et al. Genome-wide screening identifies six genes that are associated with susceptibility to Escherichia coli Microcin PDI. *Appl. Environ. Microbiol.* **81**, 6953-6963 (2015).
31. Low, B. Formation of merodiploids in matings with a class of Rec- recipient strains of Escherichia coli K12. *Proc Natl Acad Sci U S A* **60**, 160-167 (1968).
32. Tong, M. et al. Gene Dispensability in Escherichia coli Grown in Thirty Different Carbon Environments. *mBio* **11**, e02259-20 (2020).

Reviewers' Comments:

Reviewer #1:

Remarks to the Author:

The authors responded to each of the reviewers' points with a thoughtful response. Generally, the responses were satisfactory. The technical explanations were improved and easier to understand. In particular,

The results using ChIP-qPCR and fluorescence polarization assay for the verification of direct gene regulation by transcription factors, which had been uncertain, were captured and made credible (Comments #4.1, #5.4, #5.9 by this reviewer).

Controls for gene expression were properly installed and showed significance (Comment #5.3).

This study was composed in the Babu laboratory with the involvement of many researchers and the submission of much data. This reviewer pointed out details of the data that had been overlooked, which were rechecked and corrected to what they should have been and the discrepancies were resolved (Comments #5.5, #5.8, #6.2, #6.3, etc). However, these things should have been proofread in the pre-submission review. Careful confirmation by all authors themselves is needed once again before this paper is published publicly.

Ref 10 and 12 of the original manuscript were removed in response to Comment #9, but the citations were from the Ishihama group, which is one of a central research groups to the identification of the regulatory network of function unknown transcription factors. It is appropriate to cite these references as functional analysis of transcription factors in the relevant section of the introduction, and this reviewer pointed out that the method is not ChIP and should be cited as SELEX-chip in vitro as it should be.

Reviewer #2:

Remarks to the Author:

Response: We thank the reviewer for this suggestion, and have now referenced prior work²¹ for non-essential deletion mutant (alanyl-tRNA synthetase *alaS*) but not the hypomorphic essential antitoxin *mazE* that had an effect on bacterial growth in our study.

I am quite confused about this answer.

Alanyl tRNA synthetase *alaS* should be an essential gene and this essentiality is well agreed with EcoCyc, PEC, and Keio collection validation. If the authors have established as deletion mutant of this tRNA synthetase, it should be merodiploid. And the reference by Kelly et al., newly cited, reported the result using not the deletion mutant of this tRNA synthetase but 1996TGT->GCG mutation of *alaS*.

The authors should check this and make this confusion clear.

I think the responses to the rest of the comments are appropriate and acceptable for the publication of this journal except the concern above.

Reviewer #3:

Remarks to the Author:

The paper is improved and can be accepted.

Response to Reviewers

We once again thank the reviewers for providing positive feedback on our manuscript. In addition to changes to the text made based on the 'author checklist, we have addressed the remaining minor concerns of the first and second reviewer within the main text (outlined point-by-point below).

Reviewer #1:

Comment #1: The authors responded to each of the reviewers' points with a thoughtful response. Generally, the responses were satisfactory. The technical explanations were improved and easier to understand. In particular, the results using ChIP-qPCR and fluorescence polarization assay for the verification of direct gene regulation by transcription factors, which had been uncertain, were captured and made credible (Comments #4.1, #5.4, #5.9 by this reviewer). Controls for gene expression were properly installed and showed significance (Comment #5.3). This study was composed in the Babu laboratory with the involvement of many researchers and the submission of much data. This reviewer pointed out details of the data that had been overlooked, which were rechecked and corrected to what they should have been and the discrepancies were resolved (Comments #5.5, #5.8, #6.2, #6.3, etc.).

Response: We thank the reviewer for the positive feedback and acknowledging our effort.

Comment #2: Ref 10 and 12 of the original manuscript were removed in response to Comment #9, but the citations were from the Ishihama group, which is one of a central research groups to the identification of the regulatory network of function unknown transcription factors. It is appropriate to cite these references as functional analysis of transcription factors in the relevant section of the introduction, and this reviewer pointed out that the method is not ChIP and should be cited as SELEX-chip in vitro as it should be.

Response: This is an oversight on our end, and have included the references as suggested.

Reviewer #2:

Comment # 1: Alanyl tRNA synthetase *alaS* should be an essential gene and this essentiality is well agreed with EcoCyc, PEC, and Keio collection validation. If the authors have established as deletion mutant of this tRNA synthetase, it should be merodiploid. And the reference by Kelly et al., newly cited, reported the result using not the deletion mutant of this tRNA synthetase but 1996TGT->GCG mutation of *alaS*. The authors should check this and make this confusion clear.

Response: We thank the reviewer for pointing this out as alanyl-tRNA synthetase (*alaS*) Keio mutant had a partial duplication which was latter suggested to be a likely essential gene candidate (PMID: 20029369). This could be the reason why we were unable to grow this mutant strain in our experimental condition, and hence this gene was excluded from genetic screening. We have now incorporated this point in the revised text for clarity.